# How Does Message Passing Improve Collaborative Filtering?

## Abstract

Collaborative filtering (CF) has exhibited prominent results for recommender systems and is broadly utilized for real-world applications. A branch of research enhances CF methods with message passing used in graph neural networks, due to its strong capabilities of extracting knowledge from graph-structured data, like user-item bipartite graphs that naturally exist in CF. They assume that message passing helps CF methods in a manner akin to its benefits for graph-based learning tasks in general (e.g., node classification). However, whether or not this assumption is correct still needs verification, even though message passing empirically improves CF. To address this gap, we formally investigate why message passing helps CF from multiple perspectives (i.e., information passed from neighbors, additional gradients for neighbors, and individual improvement gains of subgroups w.r.t. the node degree) and show that many assumptions made by previous works are not entirely accurate. With our rigorously designed ablation studies and analyses, we discover that message passing **(i)** improves the CF performance primarily by information passed from neighbors instead of their accompanying gradients and **(ii)** usually helps low-degree nodes more than high-degree nodes. Utilizing these novel findings, we present $\underline{\text{T}}$est-time $\underline{\text{Ag}}$gregation for $\underline{\text{C}}$ollaborative $\underline{\text{F}}$iltering , namely **TAG-CF**, a test-time augmentation framework that only conducts message passing once at inference time. It can be used as a plug-and-play module and is effective at enhancing representations trained by different CF supervision signals. Evaluated on five datasets, TAG-CF performs on par with or better than trending graph-based CF methods with less than **1%** of their total training time. Furthermore, we show that test-time aggregation in TAG-CF improves recommendation performance in similar ways as the training-time aggregation does, demonstrating the legitimacy of our findings on why message passing improves CF.

## 1 Introduction

Recommender systems are essential in improving users' experiences on web services, such as product recommendations on e-commerce websites (Wang et al., 2021a; Schafer et al., 1999), video recommendations from streaming services (Gomez-Uribe & Hunt, 2015; Van den Oord et al., 2013), friend suggestions by social media platforms (Sankar et al., 2021; Ying et al., 2018), etc. In particular, recommender systems based on collaborative filtering (CF) have shown superior performance (Rendle et al., 2009; Wang et al., 2022; Koren et al., 2021). CF methods use preferences for items by users to predict additional topics or products a user might like (Su & Khoshgoftaar, 2009). These methods typically learn a unique representation for each user/item and an item is recommended to a user according to the similarity of their representations (He et al., 2017; Wang et al., 2015).

Recently, one popular line of research explores Graph Neural Networks (GNNs) for CF, exhibiting improved results compared with CF frameworks without the utilization of graphs (He et al., 2020; Wang et al., 2019; Yu et al., 2022; Cai et al., 2023). The key mechanism behind GNNs is message passing, where each node iteratively aggregates information from its direct neighbors in the graph, and information from neighbors that are multiple hops away can be acquired by stacked convolution layers (Kipf & Welling, 2016; Veličković et al., 2017; Hamilton et al., 2017). During the model training, traditional CF methods directly fetch user/item representations of an observed interaction (e.g., purchase, friending, click, etc.) and enforce their pair-wise similarity (Rendle et al., 2009). GNN-enhanced CF methods extend the aforementioned scheme by conducting stacked graph convolutions

over the user-item bipartite graph (Wang et al., 2019). They harness the resultant latent user/item representations after graph convolutions to calculate pair-wise similarity or affinity.

A recent study (He et al., 2020) shows that removing several significant components of the graph convolution (e.g., learnable transformation parameters and activation functions) greatly enhances GNNs' performance for CF. Its proposed method (LightGCN) achieves promising performance by linearly aggregating neighbor representations without any transformation, and it has been used as the de facto backbone model for later graph-based CF methods due to its simple and effective design (Cai et al., 2023; Yu et al., 2022; Wu et al., 2021). However, this observation contradicts GNN architectures for classic graph learning tasks, where GNN's performance without these components could be severely jeopardized (Oloulade et al., 2021; Wang et al., 2021b). Additionally, existing research (He et al., 2020; Wang et al., 2019) assumes that the contribution of message passing for CF is similar to that for graph learning tasks in general (e.g., node classification or link prediction - they posit that node representations are progressively refined by their neighbor information and the performance gain is positively proportional to the neighborhood density as measured in node degrees (Tang et al., 2020). However, according to our empirical studies in Section 3.2, message passing in CF improves low-degree users more than high-degree users, which also contradicts GNNs' behaviors for classic tasks (Tang et al., 2020; Hu et al., 2022). In light of these inconsistencies between the behaviors of message passing for CF and classic graph learning tasks, we ask:

*What role does message passing really play for collaborative filtering?*

In this work, we investigate contributions brought by message passing for CF from two perspectives. Firstly, we unroll the formulation of message passing and show that its performance improvement could either come from the beneficial neighbor information or additional gradient updates to neighbor representations. With rigorously designed ablation studies, we empirically demonstrate that performance gains brought by the beneficial neighbor information dominate those brought by additional gradient updates. Furthermore, we analyze the performance distribution w.r.t. the user degree (i.e., the number of interactions per user) with or without message passing and discover that the message passing in CF improves low-degree users more compared to high-degree users. For the first time, we connect this phenomenon to Laplacian matrix learning (Zhu et al., 2021; Dong et al., 2019; 2016), and show that popular supervision signals (Rendle et al., 2009; Wang et al., 2022) for CF inadvertently conduct graph convolution in the backward step even without treating the input data as a graph. Hence, when message passing is applied, high-degree users demonstrate limited improvement, as the benefit of message passage for high degree nodes has already been captured by the supervision signal.

With the above takeaways, we present **T**est-time **Ag**gregation for **C**ollaborative **F**iltering , namely **TAG-CF**. Specifically, TAG-CF does not require any message passing during training. Instead, it is a test-time augmentation framework that only conducts a single message-passing step at inference time, and effectively enhances representations inferred from different CF supervision signals. The test-time aggregation is inspired by our first perspective that, within total performance gains brought by message passing, gains from the beneficial neighbor information dominate those brought by additional gradient updates. Applying message passing only at test time avoids repetitive queries (i.e., once per node and epoch) for representations of surrounding neighbors, which grow exponentially as the number of layers increases. Moreover, following our second perspective that message passing helps low-degree nodes more in CF, we further offload the cost of TAG-CF by applying the one-time message passing only to low-degree nodes. In short, we summarize our contributions as:

- This is the first work that formally investigates why message passing helps collaborative filtering from multiple perspectives (i.e., information passed from neighbors, additional gradients for neighbors, and individual improvement gains of subgroups w.r.t. the node degree).

- With our rigorously designed ablation studies and analyses, we demonstrate that message passing in CF improves the recommendation performance primarily by information passed from neighbors instead of additional gradients, and it usually helps low-degree nodes more than high-degree nodes.

- Given our findings, we propose TAG-CF, a test-time aggregation framework effective at enhancing representations inferred by different CF supervision signals such as BPR and DirectAU. TAG-CF conducts message passing only once at test time and offers ∼4× speedup to baselines already efficient enough. Evaluated on five datasets, TAG-CF performs at par with or better than SoTA methods with a fraction of computational overhead (i.e., less than 1.0% of the total training time).

## 2 PRELIMINARY AND RELATED WORK

**Collaborative Filtering**. Given a set of users, a set of items, and interactions between users and items, collaborative filtering (CF) methods aim at learning a unique representation for each user and item, such that user and item representations can reconstruct all observable interactions (Rendle et al., 2009; Wang et al., 2022; Koren et al., 2009). CF methods based on matrix factorization directly utilize the inner product between a pair of user and item representations to infer the existence of their interaction (Koren et al., 2009; Rendle et al., 2009). Whereas CF methods based on neural predictors use multi-layer feed-forward neural networks that take user and item representations as inputs and output prediction results (He et al., 2017; Zhang et al., 2019). Let $\mathcal{U}$ and $\mathcal{I}$ denote the user set and item set respectively, with user $u_i \in \mathcal{U}$ associated with an embedding $\mathbf{u}_i \in \mathbb{R}^d$ and item $i_i \in \mathcal{U}$ associated with $\mathbf{i}_i \in \mathbb{R}^d$, the similarity $s_{ij}$ between user $u_i$ and item $i_j$ is formulated as $s_{ij} = \hat{\mathbf{u}}_i^\mathsf{T} \cdot \hat{\mathbf{i}}_j$.

**Graph Neural Networks**. Graph neural networks (GNNs) are powerful learning frameworks to extract representative information from graphs (Kipf & Welling, 2016; Veličković et al., 2017; Hamilton et al., 2017; Xu et al., 2018b). They aim at mapping each input node into low-dimensional vectors, which can be utilized to conduct either graph-level (Xu et al., 2018a) or node-level tasks (Kipf & Welling, 2016). Most GNNs explore layer-wise message passing (Gilmer et al., 2017), where each node iteratively extracts information from its first-order neighbors, and information from multi-hop neighbors can be captured by stacked layers. Given a graph $\mathcal{G} = (\mathcal{V}, \mathcal{E})$ and node features $\mathbf{X} \in \mathbb{R}^{|\mathcal{V}| \times d}$, graph convolution (Kipf & Welling, 2016) at $k$-th layer is formulated as:

$$\mathbf{h}_i^{(k+1)} = \sigma\left(\sum_{j \in \mathcal{N}(i) \cup i} \frac{1}{\sqrt{|N(i)|}\sqrt{|N(j)|}} \mathbf{h}_j^{(k)} \cdot \mathbf{W}^{(k)}\right), \tag{1}$$

where $\mathbf{h}_i^0 = \mathbf{x}_i$, $\mathcal{N}(i)$ refers to the set of direct neighbors of node $i$, and $\mathbf{W}^{(k)} \in \mathbb{R}^{d^k \times d^{(k+1)}}$ refers to parameters at the $k$-th layer transforming the node representation from $d^k$ to $d^{(k+1)}$ dimension.

Recent works (Ma et al., 2022; 2021) have shown that GNNs make predictions based on the distribution of node neighborhoods. And GNN's performance improvement for high-degree nodes better than that for low-degree nodes (Tang et al., 2020; Hu et al., 2022). They posit that node representations are progressively refined by their neighbor information and the performance gain is positively proportional to the neighborhood density as measured in node degrees.

**Message Passing for Collaborative Filtering**. Recent research tends to apply the message passing scheme in GNNs to CF (He et al., 2020; Wang et al., 2019; Pal et al., 2020). In CF, they mostly conduct message passing between user-item bipartite graphs and utilize the resultant representations to calculate user-item similarities. For instance, NGCF (Wang et al., 2019) directly migrates the message passing scheme in GNNs (similar to Equation (1)) and applies it to bipartite graphs in CF. LightGCN (He et al., 2020) simplifies NGCF (Wang et al., 2019) by removing certain components (i.e., the self-loop, learning parameters for graph convolution, and activation functions) and further improves the recommendation performance compared with NGCF. The simplified parameter-less message passing in LightGCN can be simply formatted as:

$$\mathbf{u}_i^{(k)} = \sum_{i_j \in N(u_i)} \frac{1}{\sqrt{|N(u_i)|}\sqrt{|N(i_j)|}} \mathbf{i}_j^{(k-1)}, \quad \mathbf{i}_i^{(k)} = \sum_{u_j \in N(i_i)} \frac{1}{\sqrt{|N(i_i)|}\sqrt{|N(u_j)|}} \mathbf{u}_j^{(k-1)}, \tag{2}$$

where $N(\cdot)$ refers to the set of items or users that the input interacts with, $\mathbf{u}_i^{(0)} = \mathbf{u}_i$, and $\mathbf{i}_i^{(0)} = \mathbf{i}_i$. With $K$ layers, the final user/item representations and their similarities are constructed as:

$$\hat{\mathbf{u}}_i = \frac{1}{K+1}\sum_{k=0}^{K} \mathbf{u}_i^{(k)}, \quad \hat{\mathbf{i}}_i = \frac{1}{K+1}\sum_{k=0}^{K} \mathbf{i}_i^{(k)}, \quad s_{ij} = \hat{\mathbf{u}}_i^\mathsf{T} \cdot \hat{\mathbf{i}}_j. \tag{3}$$

According to results reported in LightGCN and NGCF (He et al., 2020; Wang et al., 2019) and empirical studies we provide in this work (i.e., Table 2 and Table 5), incorporating message passing to CF methods without graphs (i.e., matrix factorization methods (Rendle et al., 2009; He et al., 2017)) can improve the recommendation performance by up to 20%. Utilizing LightGCN as the backbone model, later works try to further improve the performance by incorporating self-supervised learning signals (Lin et al., 2022; Yu et al., 2022; Cai et al., 2023). Graph-based CF methods assume that the contribution of message passing for CF is similar to that for graph learning tasks in general

(e.g., node classification or link prediction). However, whether or not this assumption is correct still needs verification, even though message passing empirically improves CF. There also exists a branch of research that aims at accelerating or simplifying message passing in CF by adding graph-based regularization terms during the training (Shen et al., 2021; Mao et al., 2021; Peng et al., 2022). While promising, they still repetitively query representations of adjacent nodes during the training.

## 3 How Does Message Passing Improve Collaborative Filtering?

In this section, we demonstrate the reason behind why message passing helps collaborative filtering from two major perspectives. The first one focuses on inductive biases brought by the message passing explored in LightGCN, the de facto backbone model for graph-based CF methods. Whereas the second perspective focuses on the performance improvement on different node subgroups w.r.t. the node degree with and without message passing.

### 3.1 Neighbor Information vs. Accompany Gradients from Message Passing

Following the definition in Equation (2), given a one-layer LightGCN[1], we unroll the calculation of the similarity $s_{ij}$ between any user $u_i$ and item $i_j$ as the following:

$$s_{ij} = \Big(\mathbf{u}_i + \sum_{i_n \in N(u_i)} \frac{1}{\sqrt{|N(u_i)|}\sqrt{|N(i_n)|}}\mathbf{i}_n\Big)^\intercal \cdot \Big(\mathbf{i}_j + \sum_{u_n \in N(i_j)} \frac{1}{\sqrt{|N(i_j)|}\sqrt{|N(u_n)|}}\mathbf{u}_n\Big)$$

$$= \mathbf{u}_i^\intercal \cdot \mathbf{i}_j + \Big(\sum_{u_n \in N(i_j)} \frac{1}{\sqrt{|N(i_j)|}\sqrt{|N(u_n)|}}\mathbf{u}_i^\intercal \cdot \mathbf{u}_n\Big) + \Big(\sum_{i_n \in N(u_i)} \frac{1}{\sqrt{|N(u_i)|}\sqrt{|N(i_n)|}}\mathbf{i}_n^\intercal \cdot \mathbf{i}_j\Big) +$$

$$\Big(\sum_{i_n \in N(u_i)} \sum_{u_n \in N(i_j)} \frac{1}{\sqrt{|N(u_i)|}\sqrt{|N(i_n)|}\sqrt{|N(i_j)|}\sqrt{|N(u_n)|}}\mathbf{i}_n^\intercal \cdot \mathbf{u}_n\Big). \tag{4}$$

With derived similarities between user-item pairs, their corresponding representations can be updated by objectives (e.g., BPR (Rendle et al., 2009) and DirectAU (Wang et al., 2022)) that enforce the pair-wise similarity between representations of user-item pairs in the training data.

CF methods without the utilization of graphs directly calculate the similarity between user and item representations with their own representations (i.e., $s_{ij} = \mathbf{u}_i^\intercal \cdot \mathbf{i}_j$), which aligns with the first term in Equation (4). Comparing it with the formulation in Equation (4), three additional similarity terms are introduced as inductive biases: similarities between users who purchase the same item (i.e., $\mathbf{u}_i^\intercal \cdot \mathbf{u}_n$), between items that share the same buyer (i.e., $\mathbf{i}_n^\intercal \cdot \mathbf{i}_j$), and between neighbors of an observed interaction (i.e., $\mathbf{i}_n^\intercal \cdot \mathbf{u}_n$). With these three additional terms from message passing, we reason that the performance improvement brought by message passing to CF methods without graph could come from **(i)** additional neighbor information (i.e., three extra terms in Equation (4)) positively correlated with user/item interactions, or **(ii)** additional gradient updates to correlated representations.

To investigate the origin of the performance improvement brought by message passing, we designed two variants of LightGCN. The first one (LightGCN$_{\text{w/o neigh. info}}$) shares the same forward and backward procedures as LightGCN during the training but does not conduct message passing during the test time. In this variant, additional gradients brought by message passing are maintained as part of the resulting model, but information from neighbors are ablated. In the second variant

Table 1: Performance of LightGCN variants.

| Method | Yelp-2018 | Gowalla | Amazon-book |
|---|---|---|---|
| | NDCG@20 | | |
| LightGCN | 6.36 | 9.88 | 8.13 |
| w/o grad. | 6.16 (3.1%↓) | 9.87 (0.1%↓) | 7.80 (4.1%↓) |
| w/o neigh. info | 4.71 (25.9%↓) | 6.95 (29.7%↓) | 6.95 (14.5%↓) |
| w/o both | 6.09 (4.2%↓) | 9.83 (0.5%↓) | 7.75 (4.7%↓) |
| | Recall@20 | | |
| LightGCN | 11.21 | 18.53 | 12.97 |
| w/o grad. | 10.87 (3.0%↓) | 18.51 (0.1%↓) | 12.81 (1.2%↓) |
| w/o neigh. info | 8.44 (24.7%↓) | 13.06 (29.5%↓) | 11.25 (13.3%↓) |
| w/o both | 10.71 (4.5%↓) | 18.42 (0.6%↓) | 12.57 (3.1%↓) |

(LightGCN$_{\text{w/o grad.}}$), the model shares the same forward pass but drops gradients for these three additional terms during the backward propagation. Besides these two variants, we also experiment on LightGCN without message passing, denoted as LightGCN$_{\text{w/o both}}$, a matrix factorization model with the same supervision signal (i.e., BPR). Implementation details are in Appendix G.

---

[1] For the simplicity of the notation, we showcase our observation with only one layer. However, since LightGCN is fully linear, the phenomenon we show also applies to variants with arbitrary layers.

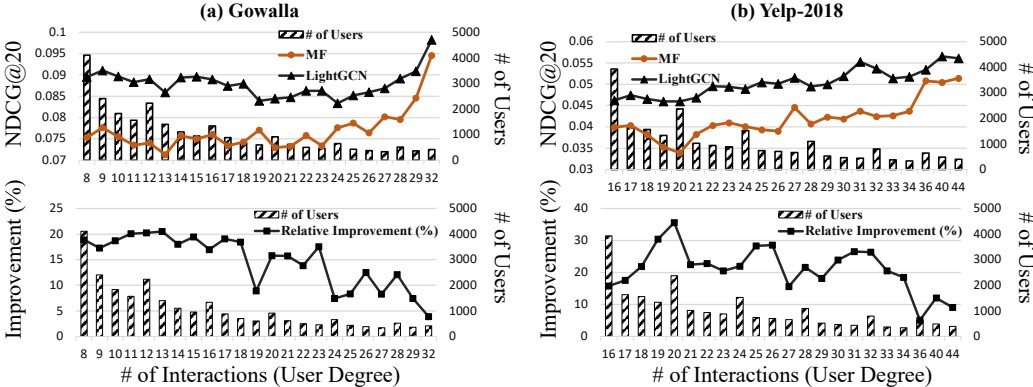

Figure 1: Performances of LightGCN and Matrix Factorization w.r.t. the user degree across datasets. The performance improvement brought by message passing decreases as the user degree goes up.

From Table 1, we observe that the performance of all variants is downgraded compared with Light-GCN, with a significant degradation on LightGCN$_{\text{w/o neigh. info}}$. This phenomenon indicates that **(i)** both neighbor information and additional gradients brought by message passing help the recommendation performance, and **(ii)** within total performance gains brought by message passing, gains from the beneficial neighbor information dominate those brought by additional gradient updates. Comparing LightGCN with LightGCN$_{\text{w/o grad.}}$, we notice that the incorporation of gradient updates brought by message passing is relatively incremental (i.e., $\sim 2\%$). However, to facilitate these additional gradient updates for slightly better performance, LightGCN is required to conduct message passing at each batch, which brings tremendous additional computational overhead.

### 3.2 MESSAGE PASSING IN CF HELPS LOW-DEGREE USERS MORE

Recent works (Tang et al., 2020; Hu et al., 2022; Liu et al., 2021) show that the performance improvement brought by GNNs is positively correlated to the number of neighbors of the nodes in question. Both empirical and theoretical evidence have demonstrated that GNNs usually perform satisfactorily on high-degree nodes with rich neighbor information but not as well on low-degree nodes. While designing graph-based model architectures for CF, most existing methods directly borrow this line of observations (Wang et al., 2019; He et al., 2020) and assume that the contribution of message passing for CF is similar to that for graph learning tasks in general. However, whether or not these observations still transfer to message passing in CF remains questionable, as there exist architectural and philosophical gaps between message passing for CF and its counterparts for GNNs, as discussed in Section 2. For instance, training signals for node classification usually come from labels agnostic of the graph structure; whereas the training signal for CF directly utilizes links between users and items to learn their representations. To validate these hypotheses, we conduct experiments over representative methods (i.e., LightGCN and matrix factorization (MF) trained with BPR) and show their performance w.r.t. the node degree in Figure 1.

We observe that, overall both MF and LightGCN perform better on high-degree users than low-degree users. According to the upper two figures in Figure 1, MF behaves similarly to LightGCN, even without treating the input data as graphs, where the overall performance for high-degree user is stronger than that for low-degree users. However, the performance improvement of LightGCN from MF on low-degree users is larger than that for high-degree users (i.e., lower two figures in Figure 1). According to literature in general graph learning tasks (Hu et al., 2022; Liu et al., 2021; Tang et al., 2020), the performance improvement should be positively proportional to the node degree - the gain for high-degree users should be higher than that for low-degree users. This discrepancy indicates that it might not be appropriate to accredit contributions of message passing in CF directly through ideologies designed for classic graph learning tasks (e.g., node classification and link prediction).

To bridge this gap, we connect supervision signals (i.e., BPR and DirectAU) commonly adopted by CF methods to Laplacian matrix learning. The formulation of BPR (Rendle et al., 2009) and

DirectAU (Wang et al., 2022) without the incorporation of graphs can be written as:

$$\mathcal{L}_{\text{BPR}} = -\sum_{(i,j)\in\mathcal{D}}\sum_{(i,k)\notin\mathcal{D}}\log\sigma(s_{ij}-s_{ik}) = -\sum_{(i,j)\in\mathcal{D}}\sum_{(i,k)\notin\mathcal{D}}\log\sigma(\mathbf{u}_i^{\mathsf{T}}\cdot\mathbf{i}_j - \mathbf{u}_i^{\mathsf{T}}\cdot\mathbf{i}_k) \qquad (5)$$

$$\mathcal{L}_{\text{DirectAU}} = \sum_{(i,j)\in\mathcal{D}}||\mathbf{u}_i - \mathbf{i}_j||^2 + \sum_{u,u'\in\mathcal{U}}\log e^{-2||\mathbf{u}-\mathbf{u}'||^2} + \sum_{i,i'\in\mathcal{I}}\log e^{-2||\mathbf{i}-\mathbf{i}'||^2}, \qquad (6)$$

where $\mathcal{D}$ refers to the set of observed interactions at the training phase and $\mathbf{i}'$ and $\mathbf{u}'$ refers to any random user/item. According to works on Laplacian matrix learning (Zhu et al., 2021; Dong et al., 2019; Ma et al., 2021), learning node representations over graphs can be decoupled into Laplacian quadratic form, a weighted summation of two sub-goals:

$$\min_{\mathbf{Z}}\{||\mathbf{Z}-\mathbf{X}||^2 + \text{tr}(\mathbf{Z}^{\mathsf{T}}\mathbf{L}\mathbf{Z})\}, \qquad (7)$$

where $\mathbf{Z}$ refers to the node representation matrix after the message passing, $\mathbf{X}$ refers to the input feature matrix, and $\mathbf{L}$ refers to the Laplacian matrix. The first term regularizes the latent representation such that it does not diverge too much from the input feature; whereas the second term promotes the similarity between latent representations of adjacent nodes, which can be re-written as: $\text{tr}(\mathbf{Z}^{\mathsf{T}}\cdot\mathbf{L}\cdot\mathbf{Z}) = \sum_{(i,j)\in\mathcal{D}}||\mathbf{u}_i - \mathbf{i}_j||^2$ in CF bipartite graphs. Zhu et al. (2021) show that $K$ layers of linear message passing exactly optimizes the second term in Equation (7). Given this theoretical foundation, we derive the following theorem w.r.t. relations between BPR, DirectAU, and message passing in CF:

**Theorem 1.** *Assuming that $||\mathbf{u}_i||^2 = ||\mathbf{i}_j||^2 = 1$ for any $u_i \in \mathcal{U}$ and $I_j \in \mathcal{I}$, objectives of BPR and DirectAU are strictly upper-bounded by the objective of message passing (i.e., $\mathcal{L}_{BPR} \leq \sum_{(i,j)\in\mathcal{D}}||\mathbf{u}_i - \mathbf{i}_j||^2$ and $\mathcal{L}_{DirectAU} \leq \sum_{(i,j)\in\mathcal{D}}||\mathbf{u}_i - \mathbf{i}_j||^2$).*

Proof of Theorem 1 can be found in Appendix A. According to Theorem 1, both BPR and DirectAU optimize the objective of message passing (i.e., $\sum_{(i,j)\in\mathcal{D}}||\mathbf{u}_i - \mathbf{i}_j||^2$) with some additional regularization (i.e., dissimilarity between non-existing user/item pairs for BPR, and representation uniformity for DirectAU). Hence, directly optimizing these two objectives partially fulfills the effects brought by message passing during the back-propagation. Combining this theoretical relation with the aforementioned empirical observations, we show that these two supervision signals could inadvertently conduct message passing in the backward step, even without explicitly treating interaction data as graphs. Since this inadvertent message passing happens during the back-propagation, its performance is positively correlated to the amount of training signals a user/item can get. In the case of CF, the amount of training signals for a user is directly proportional to the node degree. High-degree active users naturally benefit more from the inadvertent message passing from objective functions, because they acquire more training signals from the objective function. Hence, when explicit message passing is applied to CF methods, the performance gain for high-degree users is less significant than that for low-degree users. Because the contribution of the message passing over high-degree nodes has been mostly fulfilled by the inadvertent message passing during the training.

To quantitatively prove this line of theory, we incrementally upsample low-degree training examples and observe the performance improvement that TAG-CF could introduce at each upsampling rate. If our line of theory is correct, then we should expect less performance improvement on low-degree users for a larger upsampling rate. The results are shown in Appendix H.1 with supporting evidence.

## 4 TEST-TIME AGGREGATION FOR COLLABORATIVE FILTERING

In Section 3, we demonstrate why message passing helps CF from two perspectives. Firstly, w.r.t. the formulation of LightGCN, we observe that the performance gain brought by neighbor information dominates that brought by additional gradients. Secondly, w.r.t. the improvement on user subgroups, we learn that message passing helps low-degree users more, compared with high-degree users.

In light of these two takeaways, we present **T**est-time **Ag**gregation for **C**ollaborative **F**iltering, namely **TAG-CF**, a test-time augmentation framework that only conducts message passing once at inference time and is effective at enhancing matrix factorization methods trained by different CF supervision signals. Given a set of well-trained user/item representations, TAG-CF simply aggregates neighboring item (user) representations for a given user (item) at test time. Despite its simplicity,

we show that our proposal can be used as a plug-and-play module and is effective at enhancing representations trained by different CF supervision signals.

The test-time aggregation is inspired by our first perspective that, within total performance gains brought by message passing, gains from the neighbor information dominate those brought by additional gradient updates. Applying message passing only at test time avoids repetitive training-time queries (i.e., once per node and epoch) of surrounding neighbors, which grow exponentially as the number of layers increases by the neighbor explosion phenomenon (Guo et al., 2023; Zhang et al., 2021; Zeng et al., 2021). Specifically, given a set of well-trained user and item representations $\mathbf{U} \in \mathbb{R}^{|\mathcal{U}| \times d}$ and $\mathbf{I} \in \mathbb{R}^{|\mathcal{I}| \times d}$, TAG-CF augments representations for user $u_i$ and item $i_i$ as:

$$\mathbf{u}_i^* = \mathbf{u}_i + \sum_{i_j \in N(u_i)} |N(u_i)|^m |N(i_j)|^n \cdot \mathbf{i}_j, \;\; \mathbf{i}_i^* = \mathbf{i}_i + \sum_{u_j \in N(i_i)} |N(i_i)|^m |N(u_j)|^n \cdot \mathbf{u}_j, \quad (8)$$

where $m$ and $n$ are two hyper-parameters that control the normalization of message passing. With $m = n = -\frac{1}{2}$, Equation (8) becomes the exact formulation of one-layer LightGCN (i.e., Equation (2)). Empirically, we observe that the setup with $m = n = -\frac{1}{2}$ for TAG-CF does not always work for all datasets. This setup (i.e., $m = n = -\frac{1}{2}$) is directly migrated from message passing for homogeneous graphs (Kipf & Welling, 2016), which might not be applicable for bipartite graphs where all neighbors are heterogeneous (Dasoulas et al., 2021). Unlike LightGCN which can fill this gap by adaptively tuning all representations during the training time, TAG-CF cannot update any parameters since it is applied at test time, and hence requires tune-able normalization hyper-parameters.

Moreover, following our second perspective that message passing helps low-degree nodes more in CF, we further derive TAG-CF$^+$, which reduces the cost of TAG-CF by applying the one-time message passing only to low-degree nodes with sparse neighborhoods. Focusing on only low-degree nodes has two benefits: **(i)** it reduces the number of nodes that TAG-CF$^+$ needs to attend to, and **(ii)** message passing for low-degree nodes is naturally cheaper than for high-degree nodes given the surrounding neighborhoods are sparser (mitigating neighbor explosion). The degree threshold that determines which nodes to apply TAG-CF$^+$ is selected by validation performance, with details in Appendix F.

Leveraging our novel takeaways, TAG-CF can effectively enhance MF methods by conducting message passing only once at test time. It is extremely flexible, simple to implement, and enjoys the performance benefits of graph-based CF method while paying the lowest overall scalability.

## 5 EXPERIMENTS

In this section, we conduct extensive experiments to demonstrate the effectiveness and efficiency of TAG-CF. Specifically, we aim to answer the following research questions: **RQ (1)**: how effective is TAG-CF at improving MF methods without using graphs, **RQ (2)**: how much computational overhead does TAG-CF introduce, **RQ (3)**: can TAG-CF effectively enhance MF methods trained by different objectives, **RQ (4)**: how effective is TAG-CF$^+$ w.r.t. different degree cutoffs, and **RQ (5)**: do behaviors of TAG-CF align with our findings in Section 3?

### 5.1 EXPERIMENTAL SETTINGS

**Datasets**. We conduct comprehensive experiments on four commonly used benchmark datasets, including `Amazon-book`, `Anime`, `Gowalla`, and `Yelp2018`. Additionally, we also evaluate on a large-scale internal production user-item recommendation dataset `Internal`. These datasets cover different domains and dimensions to fully evaluate all models.

**Baselines**. We compare TAG-CF with two branches of methods: (1) CF methods that do not utilize graphs, including vanilla matrix factorization (MF) methods trained by BPR and DirectAU (Rendle et al., 2009; Wang et al., 2022), and Efficient Neural Matrix Factorization (Chen et al., 2020) (denoted as ENMF). (2) Graph-based CF methods, including LightGCN (He et al., 2020) and NGCF (Wang et al., 2019). Besides, we also compare with other graph-based CF methods that extend LightGCN by adding additional self-supervised signals for better performance, including LightGCL (Cai et al., 2023), SimGCL (Yu et al., 2022), and SGL (Wu et al., 2021).

Due to space limits, we include comprehensive discussions about dataset details, evaluation protocols, hyper-parameters, and other implementation details in Appendices B to E.

Table 2: Performance and running time of all models. The lower percentile indicates the set of nodes whose degrees are ranked in the lower 30% population. **Bold** and underline indicate the best and second best model respectively. LightGCN and MF are trained with DirectAU (Wang et al., 2022).

| Method | NGCF | LightGCN | ENMF | +TAG-CF | Impr. (↑) | MF | +TAG-CF | Impr. (↑%) |
|---|---|---|---|---|---|---|---|---|
| NDCG@20 – LOW-DEGREE USERS (LOWER PERCENTILE) | | | | | | | | |
| Amazon-Book | $5.32_{\pm0.08}$ | $\underline{8.09}_{\pm0.10}$ | $5.33_{\pm0.02}$ | $5.67_{\pm0.03}$ | 6.4% | $8.02_{\pm0.07}$ | $\mathbf{8.26}_{\pm0.06}$ | 3.0% |
| Anime | $20.13_{\pm0.18}$ | $\mathbf{27.78}_{\pm0.21}$ | $22.23_{\pm0.19}$ | $22.58_{\pm0.15}$ | 1.6% | $23.95_{\pm0.07}$ | $\underline{27.15}_{\pm0.04}$ | 13.4% |
| Gowalla | $8.46_{\pm0.06}$ | $\underline{10.08}_{\pm0.13}$ | $3.87_{\pm0.15}$ | $4.08_{\pm0.11}$ | 5.4% | $10.00_{\pm0.08}$ | $\mathbf{10.19}_{\pm0.04}$ | 1.9% |
| Yelp-2018 | $4.87_{\pm0.06}$ | $\underline{6.10}_{\pm0.09}$ | $3.11_{\pm0.07}$ | $3.26_{\pm0.04}$ | 4.8% | $6.08_{\pm0.08}$ | $\mathbf{6.18}_{\pm0.05}$ | 1.7% |
| Internal | $5.91_{\pm0.07}$ | $\underline{8.12}_{\pm0.03}$ | OOM | - | - | $6.79_{\pm0.04}$ | $\mathbf{8.52}_{\pm0.06}$ | 25.5% |
| NDCG@20 – OVERALL | | | | | | | | |
| Amazon-Book | $6.97_{\pm0.11}$ | $\underline{8.06}_{\pm0.11}$ | $6.13_{\pm0.13}$ | $6.54_{\pm0.09}$ | 6.7% | $8.01_{\pm0.03}$ | $\mathbf{8.13}_{\pm0.03}$ | 1.5% |
| Anime | $22.54_{\pm0.25}$ | $\mathbf{27.97}_{\pm0.21}$ | $30.17_{\pm0.09}$ | $30.86_{\pm0.12}$ | 2.3% | $24.01_{\pm0.06}$ | $\underline{27.25}_{\pm0.03}$ | 9.8% |
| Gowalla | $8.65_{\pm0.10}$ | $\mathbf{9.96}_{\pm0.11}$ | $5.23_{\pm0.04}$ | $5.29_{\pm0.05}$ | 1.1% | $9.77_{\pm0.08}$ | $\underline{9.88}_{\pm0.04}$ | 1.1% |
| Yelp-2018 | $5.54_{\pm0.06}$ | $\mathbf{6.33}_{\pm0.06}$ | $3.79_{\pm0.09}$ | $3.89_{\pm0.05}$ | 2.6% | $6.25_{\pm0.06}$ | $\underline{6.36}_{\pm0.03}$ | 1.8% |
| Internal | $6.94_{\pm0.06}$ | $\underline{8.10}_{\pm0.06}$ | OOM | - | - | $7.04_{\pm0.02}$ | $\mathbf{8.54}_{\pm0.02}$ | 21.3% |
| RECALL@20 – LOW-DEGREE USERS (LOWER PERCENTILE) | | | | | | | | |
| Amazon-Book | $10.71_{\pm0.14}$ | $\underline{13.18}_{\pm0.17}$ | $10.42_{\pm0.16}$ | $11.08_{\pm0.11}$ | 6.3% | $13.07_{\pm0.09}$ | $\mathbf{13.37}_{\pm0.10}$ | 2.3% |
| Anime | $25.74_{\pm0.35}$ | $\mathbf{32.74}_{\pm0.21}$ | $37.14_{\pm0.59}$ | $38.41_{\pm0.53}$ | 3.4% | $29.08_{\pm0.09}$ | $\underline{31.94}_{\pm0.05}$ | 9.8% |
| Gowalla | $17.53_{\pm0.32}$ | $\underline{19.14}_{\pm0.20}$ | $8.73_{\pm0.08}$ | $9.01_{\pm0.06}$ | 3.2% | $18.92_{\pm0.19}$ | $\mathbf{19.17}_{\pm0.13}$ | 1.3% |
| Yelp-2018 | $10.15_{\pm0.13}$ | $\underline{10.75}_{\pm0.14}$ | $7.17_{\pm0.06}$ | $7.54_{\pm0.12}$ | 5.2% | $10.63_{\pm0.13}$ | $\mathbf{10.98}_{\pm0.14}$ | 3.3% |
| Internal | $10.54_{\pm0.09}$ | $\underline{13.81}_{\pm0.02}$ | OOM | - | - | $11.13_{\pm0.05}$ | $\mathbf{13.97}_{\pm0.06}$ | 25.5% |
| RECALL@20 – OVERALL | | | | | | | | |
| Amazon-Book | $10.30_{\pm0.21}$ | $\underline{12.76}_{\pm0.18}$ | $10.89_{\pm0.18}$ | $11.35_{\pm0.09}$ | 4.2% | $12.67_{\pm0.06}$ | $\mathbf{12.97}_{\pm0.06}$ | 2.4% |
| Anime | $28.12_{\pm0.22}$ | $\mathbf{32.82}_{\pm0.21}$ | $34.10_{\pm0.25}$ | $34.48_{\pm0.23}$ | 1.1% | $29.15_{\pm0.09}$ | $\underline{31.95}_{\pm0.05}$ | 6.9% |
| Gowalla | $17.93_{\pm0.06}$ | $\mathbf{18.65}_{\pm0.14}$ | $9.68_{\pm0.06}$ | $9.74_{\pm0.09}$ | 0.6% | $18.30_{\pm0.17}$ | $\underline{18.53}_{\pm0.11}$ | 1.3% |
| Yelp-2018 | $10.02_{\pm0.06}$ | $\underline{10.98}_{\pm0.10}$ | $6.89_{\pm0.09}$ | $7.05_{\pm0.03}$ | 2.3% | $10.81_{\pm0.10}$ | $\mathbf{11.21}_{\pm0.09}$ | 3.7% |
| Internal | $6.91_{\pm0.04}$ | $\underline{13.89}_{\pm0.06}$ | OOM | - | - | $11.83_{\pm0.02}$ | $\mathbf{14.41}_{\pm0.08}$ | 21.8% |

Table 3: The total running time ($1 \times 10^3$ seconds) for MF methods and TAG-CF. Time % is the percentage of running time TAG-CF takes w.r.t. the time for corresponding MF methods. Speed↑ refers to the ratio of running times between training-time aggregation (i.e., LightGCN) and TAG-CF. All training steps are timed and terminated by an early stopping strategy (see Appendix E).

| Method | Sparsity | ENMF | +TAG-CF | Time % | LightGCN | MF | +TAG-CF | Time % | Speed↑ |
|---|---|---|---|---|---|---|---|---|---|
| Anime | 99.13% | 12.31 | +0.04 | 0.3% | 138.85 | 34.12 | +0.04 | 0.3% | 4.06× |
| Yelp-2018 | 99.87% | 2.15 | +0.02 | 0.9% | 5.81 | 3.17 | +0.02 | 0.6% | 1.83× |
| Gowalla | 99.91% | 4.56 | +0.02 | 0.4% | 13.27 | 7.74 | +0.02 | 0.3% | 1.72× |
| Amazon-Book | 99.94% | 11.54 | +0.03 | 0.3% | 46.62 | 29.21 | +0.03 | 0.1% | 1.59× |
| Internal | 99.99% | OOM | - | - | 47.32 | 32.62 | + 0.09 | 0.3% | 1.44× |

## 5.2 PERFORMANCE IMPROVEMENT TO MATRIX FACTORIZATION METHODS

For **RQ (1)**, Table 2 shows the performances of MF methods (MF and ENMF) as well as that of the performances of them with TAG-CF applied on their learned representations. We observe that TAG-CF unanimously improves the recommendation performance for both of them. Specifically, across all datasets, TAG-CF on average improves the low-degree NDCG@20 by 4.6% and 9.1% and overall NDCG by 3.2% and 7.1% for ENMF and MF, respectively. We also observe a similar performance improvement for Recall@20, where TAG-CF on average improves the low-degree Recall@20 by 4.5% and 8.4% and overall Recall@20 by 2.1% and 7.2% for ENMF and MF, respectively.

By comparing the performance gains brought by TAG-CF on low-degree users with that on all users, we notice that gains for low-degree users are usually higher. Hence, message passing in CF helps low-degree users more than for high-degree users. To answer **RQ (5)**, the behavior of TAG-CF aligns with our second perspective in Section 3.2 that the supervision signal inadvertently conducts message passing. Consequently, the room for improvement on high-degree users could be limited, as part of the contributions from message passing has already been claimed by the supervision signal.

## 5.3 PERFORMANCE COMPARISON AMONG GRAPH-BASED METHODS

Comparing TAG-CF with LightGCN in Table 2, we can notice that TAG-CF mostly performs on par with and sometimes even outperforms LightGCN, without incorporating message passing during the training and only conducting test-time aggregation. This phenomenon indicates that conducting neighbor aggregation at the testing time can recover most of the contributions of training-time

Table 4: The running time and performance of graph-based CF methods that extend LightGCN.

| Method | SGL | SimGCL | LightGCL | TAG-CF |
|--------|-----|--------|----------|--------|
| NDCG@20 – OVERALL | | | | |
| Anime | $27.02_{\pm 0.05}$ | $30.48_{\pm 0.12}$ | $28.34_{\pm 0.16}$ | $27.25_{\pm 0.03}$ |
| Yelp | $5.67_{\pm 0.04}$ | $5.99_{\pm 0.09}$ | $4.93_{\pm 0.06}$ | $6.36_{\pm 0.03}$ |
| Gowalla | $9.67_{\pm 0.17}$ | $10.32_{\pm 0.06}$ | $8.99_{\pm 0.13}$ | $9.88_{\pm 0.04}$ |
| Book | $6.69_{\pm 0.02}$ | $7.02_{\pm 0.05}$ | $5.83_{\pm 0.08}$ | $8.13_{\pm 0.03}$ |
| Avg. Rank | 3.2 | 1.7 | 3.5 | 1.2 |
| RECALL@20 – OVERALL | | | | |
| Anime | $31.29_{\pm 0.09}$ | $34.93_{\pm 0.14}$ | $33.64_{\pm 0.22}$ | $31.95_{\pm 0.05}$ |
| Yelp | $10.01_{\pm 0.08}$ | $10.56_{\pm 0.13}$ | $8.83_{\pm 0.04}$ | $11.21_{\pm 0.09}$ |
| Gowalla | $18.18_{\pm 0.24}$ | $19.22_{\pm 0.09}$ | $16.99_{\pm 0.10}$ | $18.53_{\pm 0.09}$ |
| Book | $11.15_{\pm 0.04}$ | $11.51_{\pm 0.09}$ | $10.06_{\pm 0.05}$ | $12.97_{\pm 0.06}$ |
| Avg. Rank | 3.2 | 1.5 | 3.5 | 1.7 |
| RUNNING TIME ($1 \times 10^3$ SECOND) | | | | |
| Anime | 69.48 | 87.77 | 97.31 | 34.15 |
| Yelp | 3.94 | 9.72 | 4.30 | 3.19 |
| Gowalla | 9.32 | 29.11 | 11.10 | 7.76 |
| Book | 63.21 | 71.39 | 38.87 | 29.24 |
| Avg. Rank | 2.2 | 3.8 | 3.0 | 1.0 |
| Total Rank | 3.6 | 2.8 | 3.9 | 1.9 |

Table 5: Performance of TAG-CF when applied to models trained with BPR loss.

| Method | LightGCN | MF | TAG-CF | Impr. (↑%) |
|--------|----------|-----|--------|------------|
| NDCG@20 – LOW-DEGREE USERS (LOWER PERCENTILE) | | | | |
| Anime | $30.02_{\pm 0.07}$ | $29.36_{\pm 0.23}$ | $30.56_{\pm 0.27}$ | 4.1% |
| Yelp | $4.34_{\pm 0.07}$ | $3.63_{\pm 0.15}$ | $3.81_{\pm 0.18}$ | 5.0% |
| Gowalla | $8.22_{\pm 0.03}$ | $7.56_{\pm 0.14}$ | $7.88_{\pm 0.15}$ | 4.2% |
| Book | $5.19_{\pm 0.14}$ | $4.19_{\pm 0.14}$ | $4.68_{\pm 0.14}$ | 11.7% |
| NDCG@20 – OVERALL | | | | |
| Anime | $30.14_{\pm 0.07}$ | $29.51_{\pm 0.21}$ | $30.23_{\pm 0.26}$ | 2.4% |
| Yelp | $4.87_{\pm 0.06}$ | $3.96_{\pm 0.14}$ | $4.26_{\pm 0.17}$ | 7.6% |
| Gowalla | $8.32_{\pm 0.03}$ | $7.51_{\pm 0.12}$ | $7.99_{\pm 0.14}$ | 6.4% |
| Book | $5.07_{\pm 0.15}$ | $4.15_{\pm 0.13}$ | $4.32_{\pm 0.13}$ | 4.1% |
| RECALL@20 – LOW-DEGREE USERS (LOWER PERCENTILE) | | | | |
| Anime | $34.23_{\pm 0.08}$ | $34.81_{\pm 0.32}$ | $35.42_{\pm 0.35}$ | 1.8% |
| Yelp | $8.19_{\pm 0.20}$ | $6.93_{\pm 0.26}$ | $7.25_{\pm 0.19}$ | 4.6% |
| Gowalla | $16.17_{\pm 0.12}$ | $14.86_{\pm 0.23}$ | $15.33_{\pm 0.24}$ | 3.2% |
| Book | $8.81_{\pm 0.26}$ | $7.45_{\pm 0.22}$ | $8.05_{\pm 0.15}$ | 8.1% |
| RECALL@20 – OVERALL | | | | |
| Anime | $34.21_{\pm 0.08}$ | $34.84_{\pm 0.30}$ | $35.23_{\pm 0.34}$ | 1.1% |
| Yelp | $8.33_{\pm 0.30}$ | $7.27_{\pm 0.27}$ | $7.62_{\pm 0.22}$ | 4.8% |
| Gowalla | $15.69_{\pm 0.07}$ | $14.47_{\pm 0.23}$ | $14.92_{\pm 0.25}$ | 3.1% |
| Book | $8.65_{\pm 0.24}$ | $7.35_{\pm 0.22}$ | $7.64_{\pm 0.20}$ | 3.9% |

message passing. To answer **RQ (5)**, TAG-CF aligns with our first perspective in Section 3.1 that the performance gain from beneficial neighbor information dominates their accompanying gradients.

We further compare TAG-CF with state-of-the-art graph-based CF methods, with their performance and efficiency shown in Table 4. Among these performant baselines, TAG-CF exhibits competitive performance, with an average rank of 1.2 on NDCG and 1.7 on Recall. Though not always the model that delivers the best performance, TAG-CF can deliver comparably promising results and introduces little computational overheads (i.e., ranked 1.0 for running time). Considering efficiency as one factor, TAG-CF achieves the best performance across all baselines and datasets with an average rank of 1.9.

While performing on par with graph-based CF methods that aggregate neighbor contents at the training time, TAG-CF enjoys the performance benefits of message passing while paying the lowest overall scalability. To answer **RQ (2)**, according to Table 3, across all datasets, TAG-CF only introduces an average additional computational overhead of $0.05 \times 10^3$ seconds, which is less than 0.5% of the total training time for matrix factorization methods. Comparing the running time of LightGCN with that of TAG-CF, we can observe that the latter can significantly improve the computational time, and the speedup is proportional to the sparsity of the dataset.

## 5.4 EFFECTIVENESS FOR DIFFERENT TRAINING SIGNALS

To answer **RQ (3)**, besides DirectAU, we also conduct experiments on BPR loss, as shown in Table 5. When applied to BPR, TAG-CF still consistently improves the performance by large margins (i.e., 6.3% and 5.1% average improvement on low-degree and overall NDCG respectively, and 4.4% and 3.2% on low-degree and overall Recall respectively). We notice that TAG-CF sometimes does not perform as competitively as LightGCN when both are trained with BPR. We check norms of learned representations from MF with BPR and discover that they have high variance since BPR does not explicitly enforce any regularization. This might not favor TAG-CF as a test-time augmentation method due to its simple design, which cannot adapt representations with high variance.

## 5.5 PERFORMANCE W.R.T. USER DEGREE

To answer **RQ (4)**, we apply TAG-CF$^+$ to four public datasets and the performance and the efficiency improvement are demonstrated in Figure 2. Overall, the running time improvement brought by TAG-CF$^+$ exponentially increases as the degree decreases, since low-degree users have sparse neighborhoods and there is hence less information for TAG-CF$^+$ to aggregation. When the degree cutoff is low (i.e., less than 100), the effectiveness of TAG-CF$^+$ proportional increases as the degree cutoff increases.

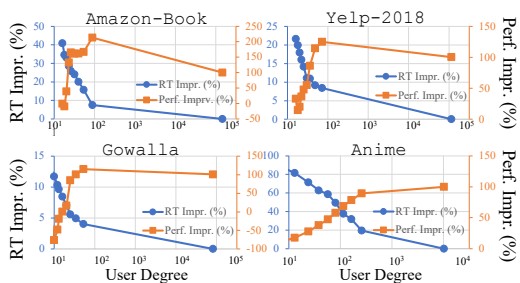

Figure 2: The performance and efficiency improvement of TAG-CF$^+$ w.r.t. different cutoffs.

When setting the cutoff to a user degree of around 100, on `Amazon-Book`, `Gowalla`, and `Yelp-2018`, TAG-CF$^+$ can further improve TAG-CF by 125%, 17%, and 11%, respectively, with efficiency improvement of 7%, 4%, and 8%. In these cases, TAG-CF$^+$ not only significantly improves the performance but also effectively reduces computational overheads. However, on these three datasets, after the cutoff bypasses a degree of 100, the performance improvement eventually decreases to the performance of TAG-CF (i.e., 100%), indicating that test-time aggregation jeopardizes the performance on high-degree nodes. On `Anime`, though no downgrade on high-degree users, the performance improvement of TAG-CF$^+$ to TAG-CF is incremental. These phenomenons not only demonstrate the effectiveness and efficiency of TAG-CF$^+$, but also verify our findings in Section 3.2 that message passing in CF helps low-degree users more than high-degree users.

## 6 Conclusion

In this study, we investigate how message passing improves collaborative filtering. Through a series of ablations, we demonstrate that the performance gain from neighbor contents dominates that from accompanying gradients brought by message passing in CF. Moreover, for the first time, we show that message passing in CF improves low-degree users more than high-degree users. We theoretically demonstrate that CF supervision signals inadvertently conduct message passing in the backward step, even without treating the data as a graph. In light of these novel takeaways, we propose TAG-CF, a test-time aggregation framework effective at enhancing representations trained by different CF supervision signals. Evaluated on five datasets, TAG-CF performs at par with SoTA methods with only a fraction of computational overhead (i.e., less than 1.0% of the total training time).

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

## A    PROOF OF THEOREM 1

Here we re-state Theorem 1 before diving into its proof:

**Theorem 1.** Assuming that $||\mathbf{u}_i||^2 = ||\mathbf{i}_j||^2 = 1$ for any $u_i \in \mathcal{U}$ and $I_j \in \mathcal{I}$, objectives of BPR and DirectAU are strictly upper-bounded by the objective of message passing (i.e., $\mathcal{L}_{\text{BPR}} \leq \sum_{(i,j)\in\mathcal{D}} ||\mathbf{u}_i - \mathbf{i}_j||^2$ and $\mathcal{L}_{\text{DirectAU}} \leq \sum_{(i,j)\in\mathcal{D}} ||\mathbf{u}_i - \mathbf{i}_j||^2$).

One preliminary theoretical foundation for Theorem 1 to hold is that a K-layer graph convolution network (GCN) exactly optimizes the second term in Equation (7), which has been proved by Zhu et al. (2021). For ease of reading, we re-phrase it again as the following:

**Theorem 2.** *The message passing for GCN optimizes the following graph regularization term:* $\mathcal{O} = \min_{\mathbf{Z}}\{tr(\mathbf{Z}^{\mathsf{T}}\mathbf{L}\mathbf{Z}))\}$.

*Proof.* Set derivative of $tr(\mathbf{Z}^{\mathsf{T}}\mathbf{L}\mathbf{Z})$ with respect to $\mathbf{Z}$ to zero:

$$\frac{\partial tr(\mathbf{Z}^{\mathsf{T}}\mathbf{L}\mathbf{Z})}{\partial \mathbf{Z}} = 0 \rightarrow \mathbf{L}\mathbf{Z} = 0 \rightarrow \mathbf{Z} = A\mathbf{Z}. \tag{9}$$

With K$\rightarrow \infty$:

$$\mathbf{Z}^{(K)} = \mathbf{A}\mathbf{Z}^{(K-1)} \tag{10}$$

which indicates:

$$\mathbf{Z}^{(K)} = \mathbf{A}\mathbf{Z}^{(K-1)} = \mathbf{A}^2\mathbf{Z}^{(K-2)} = \cdots = \mathbf{A}^K\mathbf{Z}^{(0)} = \mathbf{A}^K\mathbf{X}\mathbf{W}. \tag{11}$$

$\square$

According to this theoretical foundation, it is straightforward that Theorem 2 is also applicable for the message passing of LightGCN in the setting of CF if we let $\mathbf{A} = \{0,1\}^{(|\mathcal{U}|+|\mathcal{I}|)\times(|\mathcal{U}|+|\mathcal{I}|)}$, $\mathbf{X} = \mathbf{I}_{|\mathcal{U}|+|\mathcal{I}|}$, and $\mathbf{W} = (\mathbf{U}||\mathbf{I})$, where $||$ refers to the concatenation operation.

With these preliminaries, we can start the proof to Theorem 1 as follows:

*Proof.* DirectAU optimizes:

$$\mathcal{L}_{\text{DirectAU}} = \sum_{(i,j)\in\mathcal{D}} ||\mathbf{u}_i - \mathbf{i}_j||^2 + \sum_{u,u'\in\mathcal{U}} \log e^{-2||\mathbf{u}-\mathbf{u}'||^2} + \sum_{i,i'\in\mathcal{I}} \log e^{-2||\mathbf{i}-\mathbf{i}'||^2}. \tag{12}$$

Since $\sum_{u,u'\in\mathcal{U}} \log e^{-2||\mathbf{u}-\mathbf{u}'||^2} <= 0$ and $\sum_{i,i'\in\mathcal{I}} \log e^{-2||\mathbf{i}-\mathbf{i}'||^2} <= 0$, we directly have $\mathcal{L}_{\text{DirectAU}} \leq \sum_{(i,j)\in\mathcal{D}} ||\mathbf{u}_i - \mathbf{i}_j||^2$.

BPR optimizes:

$$\mathcal{L}_{\text{BPR}} = -\sum_{(i,j)\in\mathcal{D}} \sum_{(i,k)\notin\mathcal{D}} \log \sigma(s_{ij} - s_{ik}) = -\sum_{(i,j)\in\mathcal{D}} \sum_{(i,k)\notin\mathcal{D}} \log \sigma(\mathbf{u}_i^{\mathsf{T}} \cdot \mathbf{i}_j - \mathbf{u}_i^{\mathsf{T}} \cdot \mathbf{i}_k) \tag{13}$$

$$= \sum_{(i,j)\in\mathcal{D}} \sum_{(i,k)\notin\mathcal{D}} -\log \left(\frac{e^{\mathbf{u}_i^{\mathsf{T}} \cdot \mathbf{i}_j}}{e^{\mathbf{u}_i^{\mathsf{T}} \cdot \mathbf{i}_j} + e^{\mathbf{u}_i^{\mathsf{T}} \cdot \mathbf{i}_k}}\right) = \sum_{(i,j)\in\mathcal{D}} \sum_{(i,k)\notin\mathcal{D}} -\mathbf{u}_i^{\mathsf{T}} \cdot \mathbf{i}_j + \log \left(e^{\mathbf{u}_i^{\mathsf{T}} \cdot \mathbf{i}_j} + e^{\mathbf{u}_i^{\mathsf{T}} \cdot \mathbf{i}_k}\right)$$

$$\tag{14}$$

Since $||\mathbf{u}_i||^2 = ||\mathbf{i}_j||^2 = 1$ for any $u_i \in \mathcal{U}$ and $I_j \in \mathcal{I}$, $||\mathbf{u}_i - \mathbf{i}_j|| = \sqrt{1 - 2\mathbf{u}_i^{\mathsf{T}} \cdot \mathbf{i}_j + 1} \rightarrow -\mathbf{u}_i^{\mathsf{T}} \cdot \mathbf{i}_j = \frac{1}{2}||\mathbf{u}_i - \mathbf{i}_j||^2 - 1$. So Equation (14) can be written as:

$$\mathcal{L}_{\text{BPR}} = \frac{1}{2}||\mathbf{u}_i - \mathbf{i}_j||^2 - 1 + \log \left(e^{\mathbf{u}_i^{\mathsf{T}} \cdot \mathbf{i}_j} + e^{\mathbf{u}_i^{\mathsf{T}} \cdot \mathbf{i}_k}\right). \tag{15}$$

The maximum possible value of $e^{\mathbf{u}_i^{\mathsf{T}} \cdot \mathbf{i}_j} + e^{\mathbf{u}_i^{\mathsf{T}} \cdot \mathbf{i}_k}$ is $2e$, which is less than 10. Hence $\log \left(e^{\mathbf{u}_i^{\mathsf{T}} \cdot \mathbf{i}_j} + e^{\mathbf{u}_i^{\mathsf{T}} \cdot \mathbf{i}_k}\right) < 1$, which leads to the second part of Theorem 1: $\mathcal{L}_{\text{BPR}} \leq \sum_{(i,j)\in\mathcal{D}} ||\mathbf{u}_i - \mathbf{i}_j||^2$. $\square$

## B    DATASET DESCRIPTION AND STATISTICS

We conduct comprehensive experiments on four commonly used benchmark datasets, that have been broadly utilized by the recommender system community, including `Amazon-book` (McAuley & Yang, 2016), `Anime` (Kaggle, 2023), `Gowalla` (Cho et al., 2011), and `Yelp2018` (Yelp, 2023). Additionally, we also evaluate our method on a large-scale industrial user-content recommendation dataset - `Internal`. The detailed descriptions of these datasets are listed as the following:

- `Amazon-book`: It is a widely used product recommendation benchmark that is sub-sampled from Amazon-review[2]. In this dataset, a recommender system is asked to recommend books to users. We directly utilize the sub-sampled version created from the previous literature (Wang et al., 2022; He et al., 2020; Wang et al., 2019) to ensure a fair comparison and reproducible results. Previous works apply a 10-core setting (He & McAuley, 2016), where each user/item at least has ten interactions.

- `Anime`: It is an anime recommendation benchmark provided my MyAnimeList [3], where each user is able to add anime to their completed list and give it a rating. This dataset is a compilation of those ratings and a recommender system is asked to recommend anime to users.

- `Gowalla`: This is a check-in dataset obtained from Gowalla, where each user share locations by checking-in clicks. In this dataset, a recommender system is asked to any location that a user might be interested in checking in. Similar to `Amazon-book`, we directly utilize the processed version created from the previous literature (Wang et al., 2022; He et al., 2020; Wang et al., 2019).

- `Yelp-2018`: This dataset is adopted from the 2018 edition of the yelp challenge[4] In this dataset, a recommender system is asked to recommend businesses like local restaurants and bars to users. Similar to `Amazon-book`, we directly utilize the processed version created from the previous literature (Wang et al., 2022; He et al., 2020; Wang et al., 2019).

- `Internal`: It is a user-item interaction dataset collected from an anonymized social network platform. The dataset contains the user-item interactions from a sampled group of weekly active users of an anonymized country. The dataset contains roughly 7 million interactions, and it is split following standard train/val/test splittings.

These datasets cover different domains and dimensions to fully evaluate all models. We download `Amazon-book`, `Gowalla`, and `Yelp-2018` from the official Github repository of NGCF[5], and we acquire `Anime` from Kaggle [6]. Their statistics are shown in Table 6 below.

Table 6: Dataset Statistics. Due to privacy constrains, we only report approximated values for `Internal` dataset.

| Dataset | # Users | # Items | # Interactions | Sparsity |
|---|---|---|---|---|
| Amazon-book | 52,643 | 40,981 | 2,984,108 | 99.94% |
| Anime | 73,515 | 12,295 | 7,813,727 | 99.13% |
| Gowalla | 29,858 | 40,981 | 1,027,370 | 99.91% |
| Yelp-2018 | 31,668 | 38,048 | 1,561,406 | 99.87% |
| Internal | ~0.5M | ~0.2M | ~7M | 99.99% |

## C    EVALUATION PROTOCOL

We evaluate all models using metrics adopted in previous works, including NDCG@20 and Recall@20 (He et al., 2020). For the dataset split, we conduct the group-by-user splits and randomly select 80%, 10%, and 10% of a user's observed interactions as training, validation, and testing sets respectively. Besides, the evaluation metrics are computed by the all-ranking protocol, where all

---

[2]http://jmcauley.ucsd.edu/data/amazon.

[3]https://myanimelist.net.

[4]https://www.kaggle.com/datasets/yelp-dataset/yelp-dataset.

[5]https://github.com/xiangwang1223/neural_graph_collaborative_filtering.

[6]https://www.kaggle.com/datasets/CooperUnion/anime-recommendations-database

items are listed as candidates (Rendle et al., 2009). We explore this strategy since we want to evaluate the representation quality of all users. All experiments are conducted 10 times with different seeds, and we report both means and standard deviations across independent runs.

## D  HYPER-PARAMETER TUNING

In this section, we describe hyper-parameters that we tune for all baselines and TAG-CF and report their optimal setups. We only conduct 25 searches per model for all methods to ensure the comparison fairness, so that our experiments are not biased to methods with sophisticated hyper-parameter search spaces. Furthermore, we set the embedding dimensions for all models to 64 (i.e., $d = 64$) to ensure a fair comparison, since a larger dimension usually leads to better performance in CF methods. We utilize Adam optimizer for all models. Hyper-parameter ranges and their optimal selections are reported as follows:

- MF (BPR): We tune the learning rate from the list of [1e-4, 5e-4, 1e-3, 5e-3, 1e-2] and regularization weight (i.e., a term that regularizes the norm of learned representations) from the list of [1e-5, 1e-4, 1e-3, 1e-2, 0]. We fix the training batch size to 4096 across all datasets. he optimal selections for all datasets are listed as: `Amazon-book`: [learning rate=5e-4, regularization weight=1e-3], `Gowalla`: [learning rate=1e-3, regularization weight=0], `Yelp-2018`: [learning rate=5e-4 regularization weight=1e-5], `Anime`: [learning rate=5e-4, regularization weight=1e-4], and `Internal`: [learning rate=1e-3, regularization weight=1e-5].

- MF (DirectAU): We tune $\gamma$ (i.e., the hyper-parameter that tunes the trade-off between alignment and uniformity) from the list of $[0.1, 0.2, 0.5, 1, 2, 5, 10]$ and tune the weight decay value from the list of [0, 1e-8, 1e-6, 1e-4]. We fix the training batch size to 4096 and the learning rate to 1e-3 across all datasets. The optimal selections for all datasets are listed as: `Amazon-book`: [$\gamma = 5$, weight decay=1e-6], `Gowalla`: [$\gamma = 1$, weight decay=1e-6], `Yelp-2018`: [$\gamma = 1$, weight decay=1e-6], `Anime`: [$\gamma = 0.5$, weight decay=1e-6], and `Internal`: [$\gamma = 2$, weight decay=1e-8].

- LightGCN (BPR): We tune the learning rate from the list of [1e-4, 5e-4, 1e-3, 5e-3, 1e-2] and regularization weight (i.e., a term that regularizes the norm of learned representations) from the list of [1e-5, 1e-4, 1e-3, 1e-2, 0]. We fix the training batch size to 4096 and the number of message passing layers to 2 across all datasets. The optimal selections for all datasets are listed as: `Amazon-book`: [learning rate=1e-3, regularization weight=1e-4], `Gowalla`: [learning rate=1e-3, regularization weight=0], `Yelp-2018`: [learning rate=5e-3 regularization weight=1e-3], `Anime`: [learning rate=5e-4, regularization weight=1e-5], and `Internal`: [learning rate=5e-4, regularization weight=1e-5].

- LightGCN (DirectAU): We tune $\gamma$ (i.e., the hyper-parameter that tunes the trade-off between alignment and uniformity) from the list of $[0.1, 0.2, 0.5, 1, 2, 5, 10]$ and tune the weight decay value from the list of [0, 1e-8, 1e-6, 1e-4]. We fix the training batch size to 4096, the learning rate to 1e-3, and the number of message passing layers to 2 across all datasets. The optimal selections for all datasets are listed as: `Amazon-book`: [$\gamma = 10$, weight decay=1e-8], `Gowalla`: [$\gamma = 5$, weight decay=1e-6], `Yelp-2018`: [$\gamma = 2$, weight decay=1e-6], `Anime`: [$\gamma = 0.5$, weight decay=1e-8], and `Internal`: [$\gamma = 2$, weight decay=1e-8].

- NGCF: We tune the learning rate from the list of [1e-4, 5e-4, 1e-3, 5e-3, 1e-2], regularization weight (i.e., a term that regularizes the norm of learned representations) from the list of [1e-5, 1e-4, 1e-3, 1e-2, 0], node dropout rate [0.0, 0.1, 0.2], and message dropout rate in [0.0, 0.1, 0.2]. We fix the training batch size to 4096, learning rate to 1e-3, and the number of message passing layers to 2 across all datasets. The optimal selections for all datasets are listed as: `Amazon-book`: [learning rate=1e-3, regularization weight=1e-4, node dropout=0, message dropout=0.1], `Gowalla`: [learning rate=1e-3, regularization weight=1e-5, node dropout=0, message dropout=0.1], `Yelp-2018`: [learning rate=5e-3, regularization weight=1e-3, node dropout=0, message dropout=0], `Anime`: [learning rate=5e-4, regularization weight=1e-5, node dropout=0.1, message dropout=0.1], and `Internal`: [learning rate=5e-4, regularization weight=1e-5, node dropout=0, message dropout=0].

- ENMF: We tune dropout probability from the list of [0.3, 0.5, 0.7], learning rate from the list of [5e-3, 1e-2, 5e-2], and negative weight from the list of [0.1, 0.2, 0.5]. We fix the training batch size to 128 for `Gowalla`, `Yelp2018`, and `Anime`. For `Amazon-book`, we set the training batch size to 64. All batch sizes for ENMF is smaller compared to other baselines because of the out-of-memory issue. The optimal selections for all datasets are listed as:

`Amazon-book`:[dropout=0.3, learning rate=5e-3, negative weight=0.1], `Gowalla`:[dropout=0.5, learning rate=5e-3, negative weight=0.1], `Yelp-2018`:[dropout=0.5, learning rate=5e-3, negative weight=0.1], and `Anime`:[dropout=0.3, learning rate=5e-3, negative weight=0.5].

- LightGCL: We tune dropout probability from the list of [0.1, 0], lambda1 from the list of [0.1, 5e-3, 1e-4], lambda2 from the list of [1e-5, 1e-7], learning rate from the list of [1e-4, 1e-3, 5e-3], regularization weight from the list of [1e-5, 1e-4, 1e-3, 1e-2, 0] and temperature factor from the list of [0.2, 0.5, 0.8, 2]. We fix the training batch size to 4096 and the number of message passing layers to 2 across all datasets. The optimal selections for all datasets are listed as: `Amazon-book`:[dropout=0, lambda1=0.01, lambda2=1e-7, learning rate=1e-3, regularization weight=0.01, temperature=0.5], `Gowalla`:[dropout=0, lambda1=1e-4, lambda2=1e-7, learning rate=1e-3, regularization weight=0.001, temperature=0.5], `Yelp2018`:[dropout=0, lambda1=0.05, lambda2=1e-7, learning rate=5e-3, regularization weight=1e-5, temperature=0.5], and `Anime`:[dropout=0.1, lambda1=0.01, lambda2=1e-7, learning rate=1e-3, regularization weight=1e-5, temperature=2].

- SGL: We tune dropout probability from the list of [0.1, 0.2, 0.4, 0.5], learning rate from the list of [1e-4, 1e-3, 5e-3], regularization weight from the list of [1e-5, 1e-4, 1e-3, 1e-2, 0], temperature factor from the list of [0.1, 0.2, 0.5], and weight for the contrastive loss from the list of [0.05, 0.1, 0.5]. We fix the training batch size to 4096 and the number of message passing layers to 2 across all datasets. The optimal selections for all datasets are listed as: `Amazon-book`:[dropout=0.1, learning rate=1e-3, regularization weight=0, temperature=0.2, weight=0.5], `Gowalla`:[dropout=0.5, learning rate=1e-3, regularization weight=0.1, temperature=0.2, weight=0.5], `Yelp-2018`:[dropout=0.5, learning rate=1e-3, regularization weight=1e-4, temperature=0.2, weight=0.1], and `Anime`:[dropout=0.2, learning rate=1e-3, regularization weight=1e-5, temperature=0.5, weight=0.05].

- SimGCL: We tune epsilon from the list of [0.05, 0.1, 0.2], learning rate from the list of [1e-4, 1e-3, 5e-3], lambda from the list of [0.05, 0.01, 3e-5, 1e-5, 1e-6, 1e-7], regularization weight from the list of [1e-5, 1e-4, 1e-3, 1e-2, 0] and temperature factor from the list of [0.1, 0.2, 0.4, 0.5]. We fix the training batch size to 4096 and the number of message passing layers to 2 across all datasets. The optimal selections for all datasets are listed as: `Amazon-book`:[epsilon=0.2, learning rate=1e-3, regularization weight=1e-4, temperature=0.4, lambda=0.5], `Gowalla`:[epsilon=0.05, learning rate=1e-3, regularization weight=1e-3, temperature=0.2, lambda=1e-5], `Yelp=2018`:[epsilon=0.2, learning rate=5e-3, regularization weight=0.01, temperature=0.5, lambda=0.01], and `Anime`:[epsilon=0.05, learning rate=5e-3, regularization weight=1e-4, temperature=0.5, lambda=1e-7].

- TAG-CF (BPR): TAG-CF is a test-time augmentation framework and hence we don't tune any hyper-parameters for the model training. We only tune $m$ and $n$ in Equation (8) during test time from the list of [-2, -1.5, -1, -0.5, 0]. The optimal selections for all datasets are listed as: `Amazon-book`:[$m$=-0.5, $n$=-1], `Gowalla`:[$m$=-1.5, $n$=-0.5], `Yelp-2018`:[$m$=-1.5, $n$=-0.5], `Anime`:[$m$=-0.5, $n$=-0.5], and `Internal`:[$m$=-0.5, $n$=-0.5].

- TAG-CF (DirectAU): We only tune $m$ and $n$ in Equation (8) during test time from the list of [-2, -1.5, -1, -0.5, 0]. The optimal selections for all datasets are listed as: `Amazon-book`:[$m$=-0.5, $n$=-1], `Gowalla`:[$m$=-1.5, $n$=-0.5], `Yelp-2018`:[$m$=-1, $n$=-0.5], `Anime`:[$m$=-0.5, $n$=-0.5], and `Internal`:[$m$=-0.5, $n$=-0.5].

Besides, we adopt an early stopping strategy to train all models, where the training for a given model will be terminated if its corresponding validation NDCG@20 stops increasing for 3 continuous epochs. We use models with the best validation performance to report the performance.

## E  IMPLEMENTATION DETAIL

We conduct most of the baseline experiments with RecBole[7] (Zhao et al., 2021; 2022) (i.e., implementations of ENMF, MF, LightGCN, SimGCL, and SGL). For DirectAU, we utilize the code from its official repository[8]. We sincerely appreciate the authors of RecBole and baseline models for open-sourcing their valuable code and reliable implementations of baseline models. TAG-CF is

---

[7]https://recbole.io.
[8]https://github.com/THUwangcy/DirectAU.

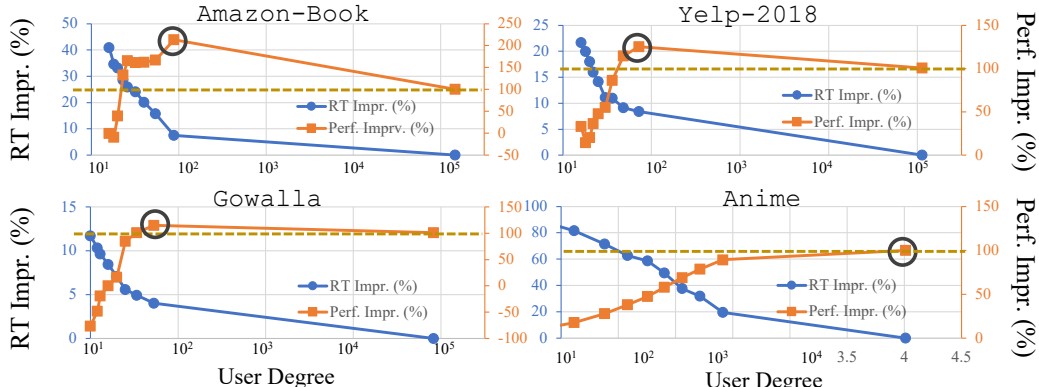

Figure 3: The performance and efficiency improvement of TAG-CF$^+$ w.r.t. different cutoffs. Yellow dashed lines indicate the performance of the regular TAG-CF, and black circles refer to the degree cutoff that TAG-CF$^+$ selects and its corresponding performance.

implemented in PyTorch 1.13 and PyG 2.2.0. As for the hardware we use to train all models, we use Google Cloud Platform with 12 CPU cores, 64GB RAM, and a single V100 GPU with 16GB VRAM to run all experiments.

## F  DEGREE CUTOFF SELECTION FOR TAG-CF$^+$

In this section, we illustrate how TAG-CF$^+$ selects the degree cutoff to improve the recommendation efficiency and effectiveness. We first sort all users according to their degree and split the sorted list into 10 user buckets[9], where each bucket contains non-overlapped users with similar degrees (e.g., the first bucket contains user whose degrees are ranked in the 0 to 10% population). Starting from the bucket with the lowest user degree, TAG-CF$^+$ keeps applying test-time-aggregation demonstrated in Equation (8) to all buckets until the validation performance starts to decrease or the performance improvement is less than 2% compared with TAG-CF. The degree cutoffs circled in Figure 3 are the ones selected by this strategy and most of them correspond to the most performant configuration. The performance of TAG-CF$^+$ under the selected degree cutoff is shown in Table 7.

Table 7: Overall Performance and efficiency improvement of TAG-CF$^+$ to TAG-CF$^+$ for different supervision signals. Degree cutoffs are selected according to circles in Figure 3.

| Metric | Yelp-2018 | Gowalla | Amazon-book | Anime |
|---|---|---|---|---|
| | BPR | | | |
| NDCG@20 | 27.1% | 10.3% | 122.4% | 0% |
| Recall@20 | 31.4% | 14.2% | 119.2% | 0% |
| Running Time | 8% | 4% | 9% | 0% |
| | DIRECTAU | | | |
| NDCG@20 | 34.1% | 22.5% | 98.3% | 0% |
| Recall@20 29.2% | 30.1% | 104.1% | 0% | |
| Running Time | 8% | 4% | 9% | 0% |

## G  IMPLEMENTATION DETAILS FOR ABLATIONS IN SECTION 3

All models in Table 1 are trained by DirectAU following the same protocols as mentioned in Appendices C and D. For LightGCN$_{\text{w/o neigh. info}}$, it shares the same forward and backward procedures as LightGCN during the training but does not conduct message passing during the test time. For LightGCN$_{\text{w/o grad.}}$, the model shares the same forward pass but drops gradients for these three additional terms during the backward propagation.

---

[9]The number of buckets can be set to arbitrary numbers for finer adjustments. In this study, we pick 10 as a proof of concept and it already effectively helps us verify the performance of TAG-CF$^+$ as well as the legitimacy of our findings.

# H ADDITIONAL EXPERIMENTS

## H.1 EXPERIMENTS ON UP-SAMPLING LOW-DEGREE NODES

In our manuscript, we connect CF objective functions (i.e., BPR and DirectAU) to message passing and show that they inadvertently conduct message passing during the back-propagation. Since this inadvertent message passing happens during the back-propagation, its performance is positively correlated to the amount of training signals a user/item can get. In the case of CF, the amount of training signals for a user is directly proportional to the node degree of this user. High-degree active users naturally benefit more from the inadvertent message passing from objective functions like BPR and DirectAU, because they acquire more training signals from the objective function. Hence, when explicit message passing is applied to CF methods, the performance gain for high-degree users is less significant than that for low-degree users. Because the contribution of the message passing over high-degree nodes has been mostly fulfilled by the inadvertent message passing during the training.

To quantitatively prove this line of theory, we incrementally upsample low-degree training examples and observe the performance improvement that TAG-CF could introduce at each upsampling rate. If our line of theory is correct, then we should expect less performance improvement on low-degree users for a larger upsampling rate. The results are shown in Table 9.

From this table, though upsampling low-degree users hurts the overall performance, we can observe that the performance improvement brought by TAG-CF for low-degree users decreases, as the upsampling rate increases. For instance, when we regard users with a degree less than 40 as low-degree users, increasing the upsampling rate from 100% to 300% reduces the improvement margin by 8.1%, with similar trends on other degree cutoffs.

According to this experiment, we can conclude that the more supervision signals a user receives (no matter for a low-degree or high-degree user), the less performance improvement message passing can bring. This experiment quantitatively shows why the performance improvement of high-degree users could be limited more than low-degree users. Because high-degree users naturally receive more training signals during the training whereas low-degree users receive fewer training signals.

Table 8: The performance improvement (NDCG@20) brought by TAG-CF at different node degree cutoffs and upsampling rates on Movielens-1M.

| Up-sampling Degree/Rate | 100% | 200% | 300% |
|---|---|---|---|
| **MF** | | | |
| 40 | 20.62 | 19.93 | 19.30 |
| 80 | 20.10 | 19.18 | 18.40 |
| 160 | 19.39 | 18.40 | 17.93 |
| **MF+TAG-CF** | | | |
| 40 | 28.87 | 26.90 | 25.01 |
| 80 | 27.43 | 24.64 | 23.30 |
| 160 | 26.63 | 24.30 | 23.37 |
| **TAG-CF IMPROVEMENT (%)** | | | |
| 40 | 38.84 | 31.2 | 30.3 |
| 80 | 35.9 | 28.2 | 26.8 |
| 160 | 36.6 | 31.8 | 29.8 |

## H.2 EXPERIMENTS ON MOVIELENS-1M

According to another recent work that uses realistic e-commerce datasets (Zheng et al., 2022), the average user degree is usually less than 50, which aligns with the characteristics of the datasets we use in this work. Nevertheless, to verify that this phenomenon is also observable in dense datasets, we apply TAG-CF to Movielens-1M. It is a dense dataset where each user has an average number of 165 interactions, as opposed to 50 interactions for other datasets that we utilize in our manuscript. The results are shown in the table below and we can notice that our observation regarding the performance improvement brought by message passing still holds.

Table 9: The performance comparison between LightGCN and TAG-CF on MovielLens-1M.

| Method | MF | +TAG-CF | Improv. over MF(%) | LightGCN | Improv. over LightGCN(%) |
|---|---|---|---|---|---|
| LOW-DEGREE PERCENTILE | | | | | |
| NDCG@20 | 20.98 | 29.20 | 39.2% | 25.95 | 12.5% |
| Recall@20 | 23.64 | 28.10 | 18.9% | 25.80 | 8.9% |
| OVERALL | | | | | |
| NDCG@20 | 22.51 | 29.65 | 31.7% | 26.64 | 11.3% |
| Recall@20 | 25.79 | 28.40 | 10.1% | 26.30 | 8.0% |

## H.3 COMPARISON EXPERIMENTS W.R.T. A SIMILAR GRAPH-BASED CF METHOD

While it might not be sensible to apply TAG-CF to graph-based CF methods that leverage message passing already, in order to fully verify our findings, we also apply TAG-CF to a graph-based method (i.e., UltraGCN (Mao et al., 2021)) that utilizes graph structures as supervision signals only. The performance as well as efficiency of UltraGCN and UltraGCN+TAG-CF are shown in Table 10.

From Table Re1, we observe that TAG-CF can further improve UltraGCN, even when the former utilizes graph structures during the model training. Specifically, TAG-CF can improve the performance of UltraGCN by 6%, which is significant. This observation indicates that the findings we propose in this manuscript can be applied to other algorithms. While TAG-CF can improve the performance, as shown in Table Re2, we can also notice that MF-DirectAU + TAG-CF in total runs a lot faster than UltraGCN. This is because UltraGCN still requires repetitive querying of the graph structures while calculating its objective functions.

Table 10: The performance (NDCG@20 and Recall@20) of UltraGCN and TAG-CF.

| Method | MF-BPR | +TAG-CF | MF-DirectAU | +TAG-CF | UltraGCN | +TAG-CF |
|---|---|---|---|---|---|---|
| OVERALL NDCG@20 | | | | | | |
| Amazon-book | $4.15_{\pm 0.13}$ | $4.32_{\pm 0.13}$ | $8.01_{\pm 0.03}$ | $8.13_{\pm 0.03}$ | $5.77_{\pm 0.25}$ | $6.11_{\pm 0.27}$ |
| Anime | $29.51_{\pm 0.21}$ | $30.23_{\pm 0.26}$ | $24.01_{\pm 0.06}$ | $27.25_{\pm 0.03}$ | $30.30_{\pm 0.11}$ | $30.89_{\pm 0.11}$ |
| Gowalla | $7.51_{\pm 0.12}$ | $7.99_{\pm 0.14}$ | $9.77_{\pm 0.08}$ | $9.88_{\pm 0.04}$ | $8.53_{\pm 0.14}$ | $9.02_{\pm 0.15}$ |
| Yelp-2018 | $3.96_{\pm 0.14}$ | $4.26_{\pm 0.17}$ | $6.25_{\pm 0.06}$ | $6.36_{\pm 0.03}$ | $5.01_{\pm 0.11}$ | $5.53_{\pm 0.11}$ |
| OVERALL RECALL@20 | | | | | | |
| Amazon-book | $7.35_{\pm 0.22}$ | $7.64_{\pm 0.20}$ | $12.67_{\pm 0.06}$ | $12.97_{\pm 0.06}$ | $8.01_{\pm 0.25}$ | $8.53_{\pm 0.25}$ |
| Anime | $34.84_{\pm 0.30}$ | $35.23_{\pm 0.34}$ | $29.15_{\pm 0.09}$ | $31.95_{\pm 0.05}$ | $35.87_{\pm 0.39}$ | $37.01_{\pm 0.39}$ |
| Gowalla | $14.47_{\pm 0.23}$ | $14.92_{\pm 0.25}$ | $18.30_{\pm 0.17}$ | $18.53_{\pm 0.11}$ | $15.93_{\pm 0.21}$ | $16.36_{\pm 0.22}$ |
| Yelp-2018 | $7.27_{\pm 0.27}$ | $7.62_{\pm 0.22}$ | $10.81_{\pm 0.10}$ | $11.21_{\pm 0.09}$ | $8.41_{\pm 0.19}$ | $9.89_{\pm 0.20}$ |

## H.4 LONGITUDINALLY CONVERTED VERSION OF TABLE 2

Below is the longitudinally converted version of Table 2 for easier comparison between the performance of low-degree and high-degree users.

## ETHICS STATEMENT

We do not observe any ethical concern entailed by our proposal, but we note that both ethical or unethical applications based on collaborative filtering may benefit from the efficiency and performance improvement of our proposal. Caution should be taken to ensure socially positive and beneficial results of representation learning algorithms. We will make sure that we follow the code of conduct and code of ethics as required by ICLR 2024.

Table 11: Performance and running time of all models. The lower percentile indicates the set of nodes whose degrees are ranked in the lower 30% population. **Bold** and underline indicate the best and second best model respectively. LightGCN and MF are trained with DirectAU (Wang et al., 2022).

| Dataset | LOW-PERCENTILE | | OVERALL | |
|---|---|---|---|---|
| | NGCG@20 | RECALL@20 | NGCG@20 | RECALL@20 |
| *Amazon-Book* | | | | |
| NGCF | $5.32_{\pm 0.08}$ | $10.71_{\pm 0.14}$ | $6.97_{\pm 0.11}$ | $10.30_{\pm 0.21}$ |
| LightGCN | $\underline{8.09}_{\pm 0.10}$ | $\underline{13.18}_{\pm 0.17}$ | $\underline{8.06}_{\pm 0.11}$ | $\underline{12.76}_{\pm 0.18}$ |
| ENMF | $5.33_{\pm 0.02}$ | $10.42_{\pm 0.16}$ | $6.13_{\pm 0.13}$ | $10.89_{\pm 0.18}$ |
| +TAG-CF | $5.67_{\pm 0.03}$ | $11.08_{\pm 0.11}$ | $6.54_{\pm 0.09}$ | $11.35_{\pm 0.09}$ |
| Impr. (↑) | 6.4% | 6.3% | 6.7% | 4.2% |
| MF | $8.02_{\pm 0.07}$ | $13.07_{\pm 0.09}$ | $8.01_{\pm 0.03}$ | $12.67_{\pm 0.06}$ |
| +TAG-CF | $\mathbf{8.26}_{\pm 0.06}$ | $\mathbf{13.37}_{\pm 0.10}$ | $\mathbf{8.13}_{\pm 0.03}$ | $\mathbf{12.97}_{\pm 0.06}$ |
| Impr. (↑) | 3.0% | 2.3% | 1.5% | 2.4% |
| *Anime* | | | | |
| NGCF | $20.13_{\pm 0.18}$ | $25.74_{\pm 0.35}$ | $22.54_{\pm 0.25}$ | $28.12_{\pm 0.22}$ |
| LightGCN | $\mathbf{27.78}_{\pm 0.21}$ | $\mathbf{32.74}_{\pm 0.21}$ | $\mathbf{27.97}_{\pm 0.21}$ | $\mathbf{32.82}_{\pm 0.21}$ |
| ENMF | $22.23_{\pm 0.19}$ | $37.14_{\pm 0.59}$ | $30.17_{\pm 0.09}$ | $34.10_{\pm 0.25}$ |
| +TAG-CF | $22.58_{\pm 0.15}$ | $38.41_{\pm 0.53}$ | $30.86_{\pm 0.12}$ | $34.48_{\pm 0.23}$ |
| Impr. (↑) | 1.6% | 3.4% | 2.3% | 1.1% |
| MF | $23.95_{\pm 0.07}$ | $29.08_{\pm 0.09}$ | $24.01_{\pm 0.06}$ | $29.15_{\pm 0.09}$ |
| +TAG-CF | $\underline{27.15}_{\pm 0.04}$ | $\underline{31.94}_{\pm 0.05}$ | $\underline{27.25}_{\pm 0.03}$ | $\underline{31.95}_{\pm 0.05}$ |
| Impr. (↑) | 13.4% | 9.8% | 9.8% | 6.9% |
| *Gowalla* | | | | |
| NGCF | $8.46_{\pm 0.06}$ | $17.53_{\pm 0.32}$ | $8.65_{\pm 0.10}$ | $17.93_{\pm 0.06}$ |
| LightGCN | $\underline{10.08}_{\pm 0.13}$ | $\underline{19.14}_{\pm 0.20}$ | $\mathbf{9.96}_{\pm 0.11}$ | $\mathbf{18.65}_{\pm 0.14}$ |
| ENMF | $3.87_{\pm 0.15}$ | $8.73_{\pm 0.08}$ | $5.23_{\pm 0.04}$ | $9.68_{\pm 0.06}$ |
| +TAG-CF | $4.08_{\pm 0.11}$ | $9.01_{\pm 0.06}$ | $5.29_{\pm 0.05}$ | $9.74_{\pm 0.09}$ |
| Impr. (↑) | 5.4% | 3.2% | 1.1% | 0.6% |
| MF | $10.00_{\pm 0.08}$ | $18.92_{\pm 0.19}$ | $9.77_{\pm 0.08}$ | $18.30_{\pm 0.17}$ |
| +TAG-CF | $\mathbf{10.19}_{\pm 0.04}$ | $\mathbf{19.17}_{\pm 0.13}$ | $\underline{9.88}_{\pm 0.04}$ | $\underline{18.53}_{\pm 0.11}$ |
| Impr. (↑) | 1.9% | 1.3% | 1.1% | 1.3% |
| *Yelp2018* | | | | |
| NGCF | $4.87_{\pm 0.06}$ | $10.15_{\pm 0.13}$ | $5.54_{\pm 0.06}$ | $10.02_{\pm 0.06}$ |
| LightGCN | $\underline{6.10}_{\pm 0.09}$ | $\underline{10.75}_{\pm 0.14}$ | $\underline{6.33}_{\pm 0.06}$ | $\underline{10.98}_{\pm 0.10}$ |
| ENMF | $3.11_{\pm 0.07}$ | $7.17_{\pm 0.06}$ | $3.79_{\pm 0.09}$ | $6.89_{\pm 0.09}$ |
| +TAG-CF | $3.26_{\pm 0.04}$ | $7.54_{\pm 0.12}$ | $3.89_{\pm 0.05}$ | $7.05_{\pm 0.03}$ |
| Impr. (↑) | 4.8% | 5.2% | 2.6% | 2.3% |
| MF | $6.08_{\pm 0.08}$ | $10.63_{\pm 0.13}$ | $6.25_{\pm 0.06}$ | $10.81_{\pm 0.10}$ |
| +TAG-CF | $\mathbf{6.18}_{\pm 0.05}$ | $\mathbf{10.98}_{\pm 0.14}$ | $\mathbf{6.36}_{\pm 0.03}$ | $\mathbf{11.21}_{\pm 0.09}$ |
| Impr. (↑) | 1.7% | 3.3% | 1.8% | 3.7% |
| *Internal* | | | | |
| NGCF | $5.91_{\pm 0.07}$ | $10.54_{\pm 0.09}$ | $6.94_{\pm 0.06}$ | $6.91_{\pm 0.04}$ |
| LightGCN | $\underline{8.12}_{\pm 0.03}$ | $\underline{13.81}_{\pm 0.02}$ | $\underline{8.10}_{\pm 0.06}$ | $\underline{13.89}_{\pm 0.06}$ |
| ENMF | OOM | | | |
| +TAG-CF | - | | | |
| Impr. (↑) | - | | | |
| MF | $6.79_{\pm 0.04}$ | $11.13_{\pm 0.05}$ | $7.04_{\pm 0.02}$ | $11.83_{\pm 0.02}$ |
| +TAG-CF | $\mathbf{8.52}_{\pm 0.06}$ | $\mathbf{13.97}_{\pm 0.06}$ | $\mathbf{8.54}_{\pm 0.02}$ | $\mathbf{14.41}_{\pm 0.08}$ |
| Impr. (↑) | 25.5% | 25.5% | 21.3% | 21.8% |

