# OpenReview forum: "How Does Message Passing Improve Collaborative Filtering?"
_ICLR.cc/2024/Conference — Submitted to ICLR 2024_

### Official Review · Reviewer_6tQK · 2023-10-29

**Soundness:** 2 fair
**Presentation:** 3 good
**Contribution:** 2 fair
**Rating:** 5
**Confidence:** 4

**Summary:**

Collaborative filtering (CF) is a widely used technique in recommender systems, and some researchers have tried to improve it using message passing in graph neural networks. However, the reasons for the improvement remain unclear. This study investigates the effects of message passing on CF from various perspectives and finds that it primarily improves CF performance through information passed from neighbors and typically benefits low-degree nodes more than high-degree ones. Based on these findings, the authors propose a test-time augmentation framework called TAG-CF, which performs message passing only once at inference time and can be used as a plug-and-play module. Tested on five datasets, TAG-CF achieves similar or better results than existing graph-based CF methods with significantly less training time. The study also shows that test-time aggregation in TAG-CF improves recommendation performance similarly to training-time aggregation, validating the findings on why message passing improves CF.

**Strengths:**

1. The ablation study of neighbor information and gradients is reasonable and novel.
2. The experiments are very detailed for reproducing; the authors provided code and configures of the experiments of information passed from neighbors, additional gradients for neighbors, and individual improvement gains of subgroups.

**Weaknesses:**

1. The analysis of chapter 3.2 is not rigorous; it doesn't safe to conclude that message passing in CF helps low-degree users More. The authors used BPR and DirectAU to denote CF methods whereas both losses belong to pair-wise learning. Generally CF also includes pointwise and listing learning, which are not discussed in the paper.
2. I agree that the Theorem 1 can reveal that for embed-based CF methods with pairwise losses: "the room for improvement on high-degree users could be limited, since part of these improvements might already have been claimed by the supervision signal itself". However, it does not mean that low-degree users will benefit more from message passing. Actually it the same conclusion for low-degree users. The difference of experimental evidences may be due to underfitting of low-degree users.
3. The authors claimed "TAG-CF cannot update any parameters since it is applied at test time, and hence requires tune-able normalization hyper-parameters". This disadvantage makes the method unstable and hard to jduge.

**Questions:**

1. Please address the concerns listed in the weakness.
2. I suggest doing up-sampling for low-degree users' data and retraining a MF or ENMF model. I conjecture this will achieve competitive result against TAG-CF variants.

---

> ### Author Response · Authors · 2023-11-18
> **Response to Reviewer 6tQK [1/4]**
>
> Dear Reviewer 6tQK:
>
> Thank you for your valuable feedback. We sincerely appreciate your acknowledgment of the novelty, comprehensiveness, as well as reproducibility of our paper. Our detailed response to your concerns is as follows:
>
> ## Weakness # 1: Analysis w.r.t. point-wise CF.
> >
> > Thanks for your comments. We would like to point out that most of the existing research that combines graphs and CF focuses on pair-wise learning schemes [1,2,3,4,5,6,7], instead of the point-wise one. For instance, all state-of-the-art graph learning methods that we compare within this work (e.g., SimGCL [1], SGL [4], LightGCL [2], and UltraGCN [3]) utilize pair-wise learning schemes. Since our paper studies graph-based CF methods, we mostly focus on pair-wise learning schemes. Besides, in our experiments, we include ENMF, a pointwise method, as a baseline model and we show that TAG-CF could still effectively and efficiently improve its performance following conclusions derived from pair-wise models. We believe that our conclusions and findings are philosophically extendable to point-wise methods and we will explore this direction as a future work.

---

> ### Author Response · Authors · 2023-11-18
> **Response to Reviewer 6tQK [2/4]**
>
> ## Weakness # 2: The difference in experimental evidences may be due to the underfitting of low-degree users.
> >
> > Thanks for your comments. In our manuscript, we connect CF objective functions (i.e., BPR and DirectAU) to message passing and show that they inadvertently conduct message passing during the back-propagation. Since this inadvertent message passing happens during the back-propagation, its performance is positively correlated to the amount of training signals a user/item can get. In the case of CF, the amount of training signals for a user is directly proportional to the node degree of this user. High-degree active users naturally benefit more from the inadvertent message passing from objective functions like BPR and DirectAU, because they acquire more training signals from the objective function. Hence, when explicit message passing is applied to CF methods, the performance gain for high-degree users is less significant than that for low-degree users. Because the contribution of the message passing over high-degree nodes has been mostly fulfilled by the inadvertent message passing during the training.
>
> > Following your suggestions, to quantitatively prove this line of theory, we incrementally upsample low-degree training examples and observe the performance improvement that TAG-CF could introduce at each upsampling rate. If our line of theory is correct, then we should expect less performance improvement on low-degree users for a larger upsampling rate. The results are shown in Table Re1.
>
> > From this table, though upsampling low-degree users hurts the overall performance, we can observe that the performance improvement brought by TAG-CF for low-degree users decreases, as the upsampling rate increases. For instance, when we regard users with a degree less than 40 as low-degree users, increasing the upsampling rate from 100% to 300% reduces the improvement margin by 8.1%, with similar trends on other degree cutoffs.
>
> > According to this experiment, we can conclude that the more supervision signals a user receives (no matter for a low-degree or high-degree user), the less performance improvement message passing can bring. This experiment quantitatively shows why the performance improvement of high-degree users could be limited more than low-degree users. Because high-degree users naturally receive more training signals during the training whereas low-degree users receive fewer training signals. However, simply adding additional supervision signals by upsampling low-degree users will hurt the overall performance. Through this series of experiments, we demonstrate the reason why the performance gain for high-degree users is less significant than that for low-degree users, when explicit message passing is applied to non-graph-based CF methods.
>
> | Upsmple Degree/ Upsample Rate |   100%   |   200%   |   300%   |
> |:----------------------------:|:-----:|:-----:|:-----:|
> | |MF|||
> |              40              | 20.62 | 19.93 | 19.30 |
> |              80              | 20.10 | 19.18 | 18.40 |
> |              160             | 19.39 | 18.40 | 17.93 |
> | |MF+TAG-CF|||
> |              40              | 28.87 | 26.90 | 25.01 |
> |              80              | 27.43 | 24.64 | 23.30 |
> |              160             | 26.63 | 24.30 | 23.37 |
> | |TAG-CF Improvement (%)|||
> |              40              | 38.4% | 31.2% | 30.3% |
> |              80              | 35.9% | 28.2% | 26.8% |
> |              160             | 36.6% | 31.8% | 29.8% |
>
> Table Re1: The performance improvement (NDCG@20) brought by TAG-CF at different node degree cutoffs and upsampling rates on Movielens-1M.

---

> ### Author Response · Authors · 2023-11-18
> **Response to Reviewer 6tQK [3/4]**
>
> ## Weakness # 3: Hyper-parameters make the methods unstable and hard to judge.
> >
> > TAG-CF only has two hyper-parameters and each of them only has 5 selections (i.e., -2, -1.5, -1, -0.5, 0), resulting in a total number of 25 runs to conduct a grid-search. Compared with TAG-CF, our baselines have a lot more hyper-parameters and each of them has a lot more selections. Moreover, tuning hyper-parameters for TAG-CF is extremely cheap as it does not induce any model retraining. For instance, as shown in Table 3, a single run of TAG-CF roughly takes less than ~1% of the total running time of an MF method. And in this case, 25 searches over the hyper-parameters only cost less than ~25% of the total running time of a single model, because TAG-CF does not affect the model training at all.  However, for all hyper-parameters in our baselines, a change in hyper-parameters would require retraining the whole model and 25 searches would cost 2500% of the total running time of a single model. We believe that the number of hyper-parameters will not void the stability of TAG-CF.

---

> ### Author Response · Authors · 2023-11-18
> **Response to Reviewer 6tQK [4/4]**
>
> ## Questions 2: Up-sampling for low-degree users' data.
> >
> > We sincerely appreciate your suggestions and the suggested upsampling experiment is really helpful for us to improve the quality of our manuscript. In our response to the second weakness that you raise, we quantitatively show that such an upsampling strategy will generally hurt the recommendation performance. However, through this series of experiments, we demonstrate the reason why the performance gain for high-degree users is less significant than that for low-degree users, when explicit message passing is applied to non-graph-based CF methods. Please refer to our response to Weakness #2 for details.
>
> ** We hope we have satisfactorily answered your questions. If so, could you please consider increasing your rating? If you have remaining doubts or concerns, please let us know, and we will happily respond.**
>
> Reference: \
> [1] Yu, Junliang, et al. "Are graph augmentations necessary? simple graph contrastive learning for recommendation." SIGIR 2022\
> [2] Cai, Xuheng, et al. "LightGCL: Simple Yet Effective Graph Contrastive Learning for Recommendation." ICLR 2023\
> [3] Kelong Mao, et al., “UltraGCN: Ultra Simplification of Graph Convolutional Networks for Recommendation” CIKM 2021\
> [4] Wu, Jiancan, et al. "Self-supervised graph learning for recommendation." SIRIR 2021\
> [5] He, Xiangnan, et al. "Lightgcn: Simplifying and powering graph convolution network for recommendation." SIGIR 2020 \
> [6] Wu, Shiwen, et al. "Graph neural networks in recommender systems: a survey." CSUR 2022\
> [7] Wang, Xiang, et al. "Neural graph collaborative filtering." SIGIR 2019 \

---

> ### Author Response · Authors · 2023-11-22
> **Reminder about Rebuttal.**
>
> Dear Reviewer 6tQK:
>
> As we conclude our rebuttal today, we look forward to the chance to interact with you. If there are any remaining questions or concerns on your end, please feel free to share them, and we'll gladly respond. Thank you.
>
> Best regards,\
> TAG-CF authors

---

### Official Review · Reviewer_gGxd · 2023-10-30

**Soundness:** 3 good
**Presentation:** 2 fair
**Contribution:** 2 fair
**Rating:** 3
**Confidence:** 4

**Summary:**

This article analyzes and empirically verifies that message passing improves collaborative filtering mainly by: (1) information passed from neighbors instead of additional gradients; (2) more benefits for low-degree nodes than high-degree nodes. Inspired by these findings, the authors propose a test-time augmentation framework that only use message passing once at inference time, named Test-time Aggregation for Collaborative Filtering (TAG-CF). And extensive experiments are conducted on five open datasets to demonstrate the effectiveness and efficiency of TAG-CF and verify the previous findings again.

**Strengths:**

1. The experiments are sufficient in the elaboration of the proposed two findings. The authors provide ablation studies on different parts of the message passing used for collaborative filtering. From the perspective of improvement gains of subgroups, the conclusions are opposite to those in other graph tasks except CF.
2. From multiple perspectives, the authors demonstrate the usability of the framework TAG-CF on CF models. Experiments are done on five datasets, especially including one large-scale dataset. And as a plug-and-play module, TAG-CF is putted in two MF methods trained by two supervision signals. Experiments are also carried out to compare time efficiency with other models.

**Weaknesses:**

1. The analysis of experiments and theories is inadequate in section 3. For example: (1) the comparison of LightGCN_(w/o both) with other variants is missing; (2) the logic of the analysis of how supervision signals can lead to limited improvement on high-degree users doesn’t make sense. The theoretical analysis that “these two supervision signals could inadvertently conduct message passing in the backward step” can’t adequately explain why the improvement on high-degree users could be limited more than low-degree users.

2. The experimental results of the proposed framework are not significant and the scope of application is narrow. (1) Although the proposed TAG-CF can improve the CF methods by using messaging in the test phase, it does not work well compared to GNN-based models, such as the mentioned SGL and LightGCL. (2) And from the design philosophy of using message passing only for testing, this framework can only be used on non-GNN-based models. However, as there are many works that modify the message passing for CF with great performance rather than being limited to LightGCN, it is difficult to bridge the gap in the benefits of training with it. (3) In terms of time efficiency, there is no significant advantage over the relatively light GNN-based models recently. You can find some more recent models to compare the efficiency to show the benefits in this regard.

3. The arrangement of Table 2 is not very reasonable. The comparison of results on low-degree users and overall performance is poorly readable. Results in these two sub-tables can be compared longitudinally in a single table.

**Questions:**

As the object you're analyzing is message passing for CF, are there any similar ablation analyses done for other Graph-based CF methods except LightGCN?

---

> ### Author Response · Authors · 2023-11-18
> **Response to Reviewer gGxd [1/4]**
>
> Dear Reviewer gGxd:
>
> Thank you for your valuable feedback. We sincerely appreciate your acknowledgment of the comprehensiveness of our experiments as well as the usability of our proposed method. Our detailed response to your concerns is as follows:
>
> ## Weakness # 1: The analysis of experiments and theories is inadequate in section 3.
> >
> > Thanks for your comments. In our manuscript, we connect CF objective functions (i.e., BPR and DirectAU) to message passing and show that they inadvertently conduct message passing during the back-propagation. Since this inadvertent message passing happens during the back-propagation, its performance is positively correlated to the amount of training signals a user/item can get. In the case of CF, the amount of training signals for a user is directly proportional to the node degree of this user. High-degree active users naturally benefit more from the inadvertent message passing from objective functions like BPR and DirectAU, because they acquire more training signals from the objective function. Hence, when explicit message passing is applied to CF methods, the performance gain for high-degree users is less significant than that for low-degree users. Because the contribution of the message passing over high-degree nodes has been mostly fulfilled by the inadvertent message passing during the training.
>
> > To quantitatively prove this line of theory, we incrementally upsample low-degree training examples and observe the performance improvement that TAG-CF could introduce at each upsampling rate. If our line of theory is correct, then we should expect less performance improvement on low-degree users for a larger upsampling rate. The results are shown in Table Re1.
>
> >From this table, though upsampling low-degree users hurts the overall performance, we can observe that the performance improvement brought by TAG-CF for low-degree users decreases, as the upsampling rate increases. For instance, when we regard users with a degree less than 40 as low-degree users, increasing the upsampling rate from 100% to 300% reduces the improvement margin by 8.1%, with similar trends on other degree cutoffs.
>
> > According to this experiment, we can conclude that the more supervision signals a user receives (no matter for a low-degree or high-degree user), the less performance improvement message passing can bring. This experiment quantitatively shows why the performance improvement of high-degree users could be limited more than low-degree users. Because high-degree users naturally receive more training signals during the training whereas low-degree users receive fewer training signals.
>
>
> | Upsmple Degree/ Upsample Rate |   100%   |   200%   |   300%   |
> |:----------------------------:|:-----:|:-----:|:-----:|
> | |MF|||
> |              40              | 20.62 | 19.93 | 19.30 |
> |              80              | 20.10 | 19.18 | 18.40 |
> |              160             | 19.39 | 18.40 | 17.93 |
> | |MF+TAG-CF|||
> |              40              | 28.87 | 26.90 | 25.01 |
> |              80              | 27.43 | 24.64 | 23.30 |
> |              160             | 26.63 | 24.30 | 23.37 |
> | |TAG-CF Improvement (%)|||
> |              40              | 38.4% | 31.2% | 30.3% |
> |              80              | 35.9% | 28.2% | 26.8% |
> |              160             | 36.6% | 31.8% | 29.8% |
>
> Table Re1: The performance improvement (NDCG@20) brought by TAG-CF at different node degree cutoffs and upsampling rates on Movielens-1M.

---

> ### Author Response · Authors · 2023-11-18
> **Response to Reviewer gGxd [2/4]**
>
> ## Weakness # 2: The experimental results of the proposed framework are not significant and the scope of application is narrow.
> >
> > **[Performance improvement]**
> The goal of TAG-CF is to match the performance of an end-to-end trained graph-based CF method (e.g., LightGCN). As shown in Table 2 in our manuscript, the performance improvement by TAG-CF is proportional to the performance gain brought by LightGCN. For instance, on Anime and Internal datasets, the performance improvement brought by TAG-CF is around 10% and 20% respectively, which is a significant improvement margin that aligns with the performance of LightGCN. However, on datasets like Gowalla and Yelp-2018, the performance improvement that could possibly be brought by the message passing scheme is incremental. Hence it will be difficult for TAG-CF to achieve further significant improvement. The goal of TAG-CF is to match the performance of graph-based CF frameworks, instead of surpassing their performance and achieving state-of-the-art performance. TAG-CF can match (and sometimes even outperform) the performance of LightGCN with only a fraction of additional computational overhead over much faster CF methods.
>
> > **[Comparison to GNN-based models]**
> Similar to the previous bullet point, the goal of TAG-CF is to efficiently match the performance of end-to-end trained graph-based CF methods, instead of surpassing their performance. While not delivering the best number in all circumstances when compared with state-of-the-art graph-based CF methods like SimGCL [1] or LightGCL [2], across these four datasets TAG-CF still achieves competitive average performance. For instance, as shown in Table 4, TAG-CF achieves an average rank of 1.2 on NDCG@20, which is the best number among all state-of-the-art models. While achieving a competitive performance, TAG-CF is also extremely efficient and simple to implement. Considering the efficiency and performance together, TAG-CF achieves the best rank among all state-of-the-art models, significantly surpassing the runner-up by 0.9 ranks. With all the aforementioned evidence, we believe that it works well compared with GNN-based models.
>
> > **[Applicability to other methods]**
> We use the message passing scheme in LightGCN to derive our findings and further propose TAG-CF. Though our conclusions are based on LightGCN, it does not limit the applicability and potential extensions of our proposal. Our research does not aim at further improving the performance of graph-based CF methods. Instead, we focus on enhancing the performance of non-graph-based CF methods in an extremely efficient and simple manner. Hence, our findings and conclusions are applicable to any CF methods without graphs. We leverage our findings and conclusions and further propose TAG-CF, which is simple to implement and does not require any architectural modifications due to its test-time augmentation trait. We believe that the generality and intuitiveness of our findings will be impactful for both academic researchers and industrial practitioners.
>
> > **[TAG-CF further improves graph-based CF very efficiently]**
> While it might not be sensible to apply TAG-CF to graph-based CF methods that leverage message passing already, in order to fully verify our findings, we also apply TAG-CF to a graph-based method (i.e., UltraGCN [6]) that utilizes graph structures as supervision signals only. The performance as well as efficiency of UltraGCN and UltraGCN+TAG-CF are shown in the table below.
> >
> > From Table Re1, we observe that TAG-CF can further improve UltraGCN, even when the former utilizes graph structures during the model training. Specifically, TAG-CF can improve the performance of UltraGCN by ~6%, which is significant. This observation indicates that the findings we propose in this manuscript can be applied to other algorithms. While TAG-CF can improve the performance, as shown in Table Re2, we can also notice that MF-DirectAU + TAG-CF in total runs a lot faster than UltraGCN. This is because UltraGCN still requires repetitive querying of the graph structures while calculating its objective functions.

---

> ### Author Response · Authors · 2023-11-18
> **Response to Reviewer gGxd [3/4]**
>
> ## Weakness # 2: The experimental results of the proposed framework are not significant and the scope of application is narrow (Cont.).
> >
> > **[Efficiency]**
> TAG-CF significantly reduces the computational overheads and conducts these operations only once for all nodes. TAG-CF+ further improves and conducts these operations only once for low-degree nodes. Besides, TAG-CF and TAG-CF+ only require 1-hop neighbor information, unlike existing works that usually require neighbors from 2 to 3 neighbors. The price that TAG-CF and TAG-CF+ pay is the minimal price if one wants to incorporate message passing of any sort in a CF system.  Compared with UltraGCN, a recently proposed efficient graph-based GNN that also does not utilize message passing during training, TAG-CF consistently runs faster, as shown in Table Re3. This is because UltraGCN still requires repetitive querying of the graph structures while calculating its objective functions.
>
> >**[Potential production impact]**
> While end-to-end trained graph-based CF methods consistently deliver promising performance, they are rarely used in real-world production scenarios, due to training complexities and prohibitively expensive overheads [8,9,10]. TAG-CF offers an extremely simple yet effective and practical approach for industry settings to benefit from the power of message passing while paying minimal cost. We believe that the theoretical and empirical findings we provide in this paper will be valuable to researchers from both academia and industry.
>
> |    Method   |   MF-BPR   | MF-BPR + TAG-CF |    MF-DirectAU    | MF-DirectAU + TAG-CF |  UltraGCN  | UltraGCN + TAG-CF |
> |:-----------:|:----------:|:---------------:|:-----------------:|:--------------------:|:----------:|:-----------------:|
> |             |            |                 |  Overall NDCG@20  |                      |            |                   |
> | Amazon-Book |  4.15±0.13 |    4.32±0.13    |     8.01±0.03     |       8.13±0.03      |  5.77±0.25 |     6.11±0.27     |
> |    Anime    | 29.51±0.21 |    30.23±0.26   |     24.01±0.06    |      27.25±0.03      | 30.30±0.11 |     30.89±0.11    |
> |   Gowalla   |  7.51±0.12 |    7.99±0.14    |     9.77±0.08     |       9.88±0.04      |  8.53±0.14 |     9.02±0.15     |
> |  Yelp-2018  |  3.96±0.14 |    4.26±0.17    |     6.25±0.06     |       6.36±0.03      |  5.01±0.11 |     5.53±0.11     |
> |             |            |                 | Overall Recall@20 |                      |            |                   |
> | Amazon-Book |  7.35±0.22 |    7.64±0.20    |     12.67±0.06    |      12.97±0.06      |  8.01±0.25 |     8.53±0.25     |
> |    Anime    | 34.84±0.30 |    35.23±0.34   |     29.15±0.09    |      31.95±0.05      | 35.87±0.39 |     37.01±0.39    |
> |   Gowalla   | 14.47±0.23 |    14.92±0.25   |     18.30±0.17    |      18.53±0.11      | 15.93±0.21 |     16.36±0.22    |
> |  Yelp-2018  |  7.27±0.27 |    7.62±0.22    |     10.81±0.10    |      11.21±0.09      |  8.41±0.19 |     9.89±0.20     |
>
> Table Re2: The performance (NDCG@20 and Recall@20) of UltraGCN and TAG-CF.
>
> |   Method    | LightGCN |   MF  | +TAG-CF | UltraGCN | +TAG-CF |
> |:-----------:|:--------:|:-----:|:-------:|:--------:|:-------:|
> |    Anime    |  138.85  | 34.12 |  +0.04  |   93.31  |  +0.04  |
> |  Yelp-2018  |   5.81   |  3.17 |  +0.02  |   5.02   |  +0.02  |
> |   Gowalla   |   13.27  |  7.74 |  +0.02  |   12.55  |  +0.02  |
> | Amazon-Book |   11.54  | 29.21 |  +0.03  |   39.26  |  +0.03  |
>
> Table Re3: The running time comparison (1 x 10^3 seconds) between TAG-CF and UltraGCN.

---

> ### Author Response · Authors · 2023-11-18
> **Response to Reviewer gGxd [4/4]**
>
> ## Weakness #3: Table 2 Arrangement
> >
> > Thanks for pointing this out. We accordingly added a modified version of Table 2 and put it in the appendix (i.e., Appendix H.4 Table 11). Upon your approval, we will add it to the main manuscript.
>
> ## Question 1: Message passing scheme beyond LightGCN
> >
> > LightGCN is broadly explored and researched in the community of recommender systems and it is used as the backbone model for state-of-the-art recommender systems such as LightGCL, SimGCL, SGL, etc. There do not exist too many different message passing schemes in this community. We believe that the message passing scheme that we explore in this manuscript (i.e., linear neighbor aggregation) is predominant and widely accepted [1,2,3,4,5,6]. The other predominant scheme is NGCF which utilizes non-linear parameterized message passing. However, compared with NGCF, LightGCN has shown that linear neighbor aggregation is more efficient and effective. However, compared with NGCF[7], LightGCN has shown that linear neighbor aggregation is more efficient and effective. Hence, the conclusions and findings we have derived in this work can be widely applied to any graph-based CF methods that explore LightGCN as the backbone model, which accounts for the vast majority of graph-based CF methods.
>
> **We hope we have satisfactorily answered your questions. If so, could you please consider increasing your rating? If you have remaining doubts or concerns, please let us know, and we will happily respond.**
>
> Reference: \
> [1] Yu, Junliang, et al. "Are graph augmentations necessary? simple graph contrastive learning for recommendation." SIGIR 2022\
> [2] Cai, Xuheng, et al. "LightGCL: Simple Yet Effective Graph Contrastive Learning for Recommendation." ICLR 2023\
> [3] Kelong Mao, et al., “UltraGCN: Ultra Simplification of Graph Convolutional Networks for Recommendation” CIKM 2021\
> [4] Wu, Jiancan, et al. "Self-supervised graph learning for recommendation." SIRIR 2021\
> [5] He, Xiangnan, et al. "Lightgcn: Simplifying and powering graph convolution network for recommendation." SIGIR 2020\
> [6] Wu, Shiwen, et al. "Graph neural networks in recommender systems: a survey." CSUR 2022\
> [7] Wang, Xiang, et al. "Neural graph collaborative filtering." SIGIR 2019\
> [8] Si, Si, et al. "Serving Graph Compression for Graph Neural Networks." ICLR 2023\
> [9] Zhang, Shichang, et al. "Graph-less neural networks: Teaching old mlps new tricks via distillation." ICLR 2022\
> [10] Guo, Zhichun, et al. "Linkless link prediction via relational distillation." ICML 2023

---

> ### Author Response · Authors · 2023-11-22
> **Reminder about Rebuttal.**
>
> Dear Reviewer gGxd:
>
> As we wrap up our discussion today, we eagerly anticipate the opportunity to engage with you. Should you have any remaining questions or concerns, please don't hesitate to share them, and we'll be more than happy to assist. Thank you.
>
> Best regards,\
> TAG-CF authors

---

### Official Review · Reviewer_uTpx · 2023-10-30

**Soundness:** 2 fair
**Presentation:** 2 fair
**Contribution:** 1 poor
**Rating:** 3
**Confidence:** 5

**Summary:**

The authors initially conduct experiments to illustrate that the message-passing process contributes more significant benefits to collaborative filtering than those derived from gradient updates. Subsequently, they discover that the message-passing mechanism offers more substantial advantages for nodes with fewer connections. Ultimately, the paper suggests that TAG-CF does not implement message-passing during the training phase. Instead, it exclusively applies this technique during the inference process.

**Strengths:**

1.	The paper attempts to explain the role of message passing in collaborative filtering, which is a promising avenue of investigation.
2.	The article is easy to read, and the tables and illustrations are quite clear and comprehensible.

**Weaknesses:**

Weaknesses:

1. The conclusions drawn from the experiments on LightGCN are well-known and lack a deeper theoretical foundation.

In Section 3.1 of the paper, the conclusions drawn are not groundbreaking; instead, they represent widely accepted knowledge, lacking innovation. Moreover, the article fails to provide theoretical underpinnings for the hypotheses and conclusions presented, making it challenging to extend these findings to collaborative filtering algorithms beyond LightGCN. The research solely relies on the LightGCN model in experiments conducted on three datasets with relatively high sparsity, which may introduce bias. The experimental results indicate two key points: 1. Message passing and gradient updates have practical significance for collaborative filtering recommendation systems, and 2. Message passing primarily contributes to performance improvements in collaborative filtering. It's worth noting that since the article exclusively investigates the LightGCN model, the conclusion predominantly emphasizes the role of message passing in enhancing LightGCN's performance. While this conclusion holds for LightGCN, it may not necessarily apply to other models, warranting further exploration.

Indeed, further theoretical exploration may be necessary.

2. The setup of the exploratory experiments is problematic, and this experimental design lacks fairness and equity.

Furthermore, in the exploratory experiments of Section 3.1, the experimental design lacks the necessary rigor to effectively validate the author's hypotheses. In the case of (LightGCNw/o neigh. info), a message passing mechanism is employed during training but not during inference. During training, embedding representations are improved through message passing to obtain new user and item embeddings, which are then used to compute BPR loss. The optimization objective of message passing is to enhance the similarity and dissimilarity between embedding features after message passing. However, in the inference phase, the experiments omit message passing and use the original features as embeddings for users and items. This results in inconsistent optimization objectives between training and inference, with inference embeddings notably lacking in similarity and collaborative signals. Consequently, the (LightGCNw/o neigh. info) model's performance unavoidably becomes suboptimal. These experimental results, therefore, cannot effectively prove that message passing is the most critical factor.

3. Applying TAG-CF solely to MF and ENMF is insufficient to demonstrate the effectiveness of TAG-CF, and in some experiments, the results show non-statistically significant improvements.

The proposed TAG-CF essentially involves deactivating message passing during model training and enabling it during inference. In the primary experiment, only ENMF and MF had an additional message passing step during inference to validate its effectiveness. However, these experiments are considered insufficiently comprehensive. It might be necessary to apply TAG-CF to a broader range of classic Graph-based recommendation models to thoroughly confirm its effectiveness. Limiting the application to the simplest ENMF and MF models alone may not be sufficient to fully demonstrate the effectiveness of TAG-CF. The comparative models solely comprise classic NGCF and LightGCN; thus, it might be necessary to include more recent models in the experiments. Additionally, on some datasets, the improvement is less than 2%, which is not statistically significant.

4. The choice of datasets in this study is limited to a single type.

Additionally, the choice of datasets in this study is confined to a single type. The paper's model primarily focuses on node degrees, but the datasets used in the experiments all have relatively high sparsity. It would be beneficial to include datasets with lower sparsity, such as Movielens-1M, to ensure a more comprehensive evaluation of the model's performance.

5. Despite the computational complexity, the improvements achieved with TAG-CF+ are not significantly greater than those of TAG-CF.

TAG-CF+ involves exclusively passing messages to nodes with lower degrees. However, the computation of node degrees, selection of low-degree nodes, and the search for neighbors of low-degree nodes represent computationally complex tasks.

**Questions:**

1. In the experiments in Section 3.2, how many layers of LightGCN are used?
2. Regarding sensitivity to node degrees, are there experiments being conducted on datasets with even higher sparsity levels, which are commonly known to have higher degrees of sparsity?

---

> ### Author Response · Authors · 2023-11-18
> **Response to Reviewer uTpx [1/7]**
>
> Dear Reviewer uTpx:
>
> Thank you for your valuable feedback. We sincerely appreciate your acknowledgment of the importance of our research as well as the clarity of our manuscript. Our detailed response to your concerns is as follows:
>
> ## Weakness # 1: The conclusions drawn from the experiments on LightGCN are well-known and lack a deeper theoretical foundation.
> >
> > **[Extension beyond LightGCN]**
> LightGCN is broadly explored and researched in the community of recommender systems and it is used as the backbone model for state-of-the-art recommender systems such as LightGCL, SimGCL, SGL, etc. There do not exist too many different message passing schemes in this community. We believe that the message passing scheme that we explore in this manuscript (i.e., linear neighbor aggregation) is predominant and widely accepted [1,2,3,4,5,6]. The other predominant scheme is NGCF which utilizes non-linear parameterized message passing. However, compared with NGCF, LightGCN has shown that linear neighbor aggregation is more efficient and effective.
> >
> > Hence, the conclusions and findings we have derived in this work can be widely applied to any graph-based CF methods that explore LightGCN as the backbone model, and CF methods based on LightGCN account for the vast majority of graph-based CF methods.
>
> > **[Applicability to other methods]**
> We use the message passing scheme in LightGCN to derive our findings and further propose TAG-CF. Though our conclusions are based on LightGCN, it does not limit the applicability and potential extensions of our proposal. Our research does not aim at further improving the performance of graph-based CF methods. Instead, we focus on enhancing the performance of non-graph-based CF methods in an extremely efficient and simple manner such that they can effectively match the performance of graph-based methods. Hence, our findings and conclusions are applicable to any CF methods without graphs. We leverage our findings and conclusions and further propose TAG-CF, which is simple to implement and does not require any training-time architectural modifications due to its test-time augmentation trait. We believe that the generality and intuitiveness of our findings will be impactful for both academic researchers and industrial practitioners.
>
> >**[TAG-CF further improves graph-based CF very efficiently]**
> While it might not be sensible to apply TAG-CF to graph-based CF methods that leverage message passing already, in order to fully verify our findings, we also apply TAG-CF to a graph-based method (i.e., UltraGCN [6]) that utilizes graph structures as supervision signals only. The performance as well as efficiency of UltraGCN and UltraGCN+TAG-CF are shown in the table below.
>
> > From Table Re1, we observe that TAG-CF can further improve UltraGCN, even when the former utilizes graph structures during the model training. Specifically, TAG-CF can improve the performance of UltraGCN by ~6%, which is significant. This observation indicates that the findings we propose in this manuscript can be applied to other algorithms. While TAG-CF can improve the performance, as shown in Table Re2, we can also notice that MF-DirectAU + TAG-CF in total runs a lot faster than UltraGCN. This is because UltraGCN still requires repetitive querying of the graph structures while calculating its objective functions.
>
> |    Method   |   MF-BPR   | MF-BPR + TAG-CF |    MF-DirectAU    | MF-DirectAU + TAG-CF |  UltraGCN  | UltraGCN + TAG-CF |
> |:-----------:|:----------:|:---------------:|:-----------------:|:--------------------:|:----------:|:-----------------:|
> |             |            |                 |  Overall NDCG@20  |                      |            |                   |
> | Amazon-Book |  4.15±0.13 |    4.32±0.13    |     8.01±0.03     |       8.13±0.03      |  5.77±0.25 |     6.11±0.27     |
> |    Anime    | 29.51±0.21 |    30.23±0.26   |     24.01±0.06    |      27.25±0.03      | 30.30±0.11 |     30.89±0.11    |
> |   Gowalla   |  7.51±0.12 |    7.99±0.14    |     9.77±0.08     |       9.88±0.04      |  8.53±0.14 |     9.02±0.15     |
> |  Yelp-2018  |  3.96±0.14 |    4.26±0.17    |     6.25±0.06     |       6.36±0.03      |  5.01±0.11 |     5.53±0.11     |
> |             |            |                 | Overall Recall@20 |                      |            |                   |
> | Amazon-Book |  7.35±0.22 |    7.64±0.20    |     12.67±0.06    |      12.97±0.06      |  8.01±0.25 |     8.53±0.25     |
> |    Anime    | 34.84±0.30 |    35.23±0.34   |     29.15±0.09    |      31.95±0.05      | 35.87±0.39 |     37.01±0.39    |
> |   Gowalla   | 14.47±0.23 |    14.92±0.25   |     18.30±0.17    |      18.53±0.11      | 15.93±0.21 |     16.36±0.22    |
> |  Yelp-2018  |  7.27±0.27 |    7.62±0.22    |     10.81±0.10    |      11.21±0.09      |  8.41±0.19 |     9.89±0.20     |
>
> Table Re1: The performance (NDCG@20 and Recall@20) of UltraGCN and TAG-CF.

---

> ### Author Response · Authors · 2023-11-18
> **Response to Reviewer uTpx [2/7]**
>
> ## Weakness # 1: The conclusions drawn from the experiments on LightGCN are well-known and lack a deeper theoretical foundation. (Cont.)
>
> |   Method    | LightGCN |   MF  | +TAG-CF | UltraGCN | +TAG-CF |
> |:-----------:|:--------:|:-----:|:-------:|:--------:|:-------:|
> |    Anime    |  138.85  | 34.12 |  +0.04  |   93.31  |  +0.04  |
> |  Yelp-2018  |   5.81   |  3.17 |  +0.02  |   5.02   |  +0.02  |
> |   Gowalla   |   13.27  |  7.74 |  +0.02  |   12.55  |  +0.02  |
> | Amazon-Book |   11.54  | 29.21 |  +0.03  |   39.26  |  +0.03  |
>
> Table Re2: The running time comparison (1 x 10^3 seconds) between TAG-CF and UltraGCN.
>
> >**[Conclusions from the experiments are well-known]**
> The most significant conclusions we propose in this work are (1) within the message passing scheme for CF, numerical values of neighbor information are more important than the accompanying gradients, and (2) explicit message passing in CF helps low-degree users more than it does for high-degree users. Based on these two findings, we propose TAG-CF, an extremely efficient test-time aggregation framework for CF. It enables CF systems without graphs to easily match the performance of graph-based CF methods. To our knowledge, these conclusions are novel and valuable to both academic researchers and industrial practitioners. If the reviewer has seen other related work with these conclusions, we are more than happy to discuss and accordingly revise our manuscript.

---

> ### Author Response · Authors · 2023-11-18
> **Response to Reviewer uTpx [3/7]**
>
> ## Weakness # 2: The setup of the exploratory experiments is problematic, and this experimental design lacks fairness and equity.
>
> >Thanks for your comment -- we do not believe the fairness of the evaluation is a concern in these experiments.   These experiments simply aim to compare two different components brought by the message-passing mechanism (i.e., neighbor information vs. accompanying gradients). Through these two variants, we aim at answering how much contribution each component can bring by ablating the other one.
>
> >Moreover, even if we remove the second variant (i.e., LightGCN without neig. info), the findings and conclusions we draw in this section mostly still hold. This is because we just want to demonstrate that the message passing in CF can still deliver good performance even when it is not involved in the back-propagation.

---

> ### Author Response · Authors · 2023-11-18
> **Response to Reviewer uTpx [4/7]**
>
> ## Weakness # 3: Applying TAG-CF solely to MF and ENMF is insufficient to demonstrate the effectiveness of TAG-CF, and in some experiments, the results show non-statistically significant improvements.
> >
> > **[Experiments insufficient to demonstrate the effectiveness of TAG-CF]**
> Thanks for pointing this out. While designing our experiments, we aim to evaluate the effectiveness of TAG-CF for enhancing the performance of non-graph-based CF methods, and the most predominant methods are MF and NMF. To alleviate your concerns,  we apply TAG-CF to an additional graph-based method (i.e., UltraGCN [6]) which utilizes graph structures as supervision signals only. The performance as well as efficiency of UltraGCN and UltraGCN+TAG-CF is shown in Table Re1.
>
> > From Table Re1, we observe that TAG-CF can further improve UltraGCN, even when the former utilizes graph structures during the model training via supervision. Specifically, TAG-CF can improve the performance of UltraGCN by ~6%, which is significant. This observation indicates that the findings we propose in this manuscript can be applied to other algorithms: namely, that explicit message passing information during forward is more valuable than the gradients in backward). While TAG-CF can improve the performance, as shown in Table Re2,  we can also notice that MF-DirectAU + TAG-CF in total runs a lot faster than UltraGCN, demonstrating both the efficiency and effectiveness of TAG-CF.
>
> > **[Performance improvement]**
> The goal of TAG-CF is to match the performance of an end-to-end trained graph-based CF method (e.g., LightGCN). As shown in Table 2 in our manuscript, the performance improvement by TAG-CF is proportional to the performance gain brought by LightGCN. For instance, on Anime and Internal datasets, the performance improvement brought by TAG-CF is around 10% and 20% respectively, which is a significant improvement margin that aligns with the performance of LightGCN. However, on datasets like Gowalla and Yelp-2018, the performance improvement that could possibly be brought by the message passing scheme is incremental. Hence it will be difficult for TAG-CF to achieve further significant improvement. The goal of TAG-CF is to match the performance of graph-based CF frameworks, instead of surpassing their performance and achieving state-of-the-art performance. TAG-CF can match (and sometimes even outperform) the performance of LightGCN with only a fraction of additional computational overhead over much faster CF methods.
>
> > **[Potential production impact]**
> While end-to-end trained graph-based CF methods consistently deliver promising performance, they are rarely used in real-world production scenarios, due to training complexities and prohibitively expensive overheads [7,8,9]. TAG-CF offers an extremely simple yet effective and practical approach for industry settings to benefit from the power of message passing while paying minimal cost. We believe that the theoretical and empirical findings we provide in this paper will be valuable to researchers from both academia and industry.

---

> ### Author Response · Authors · 2023-11-18
> **Response to Reviewer uTpx [5/7]**
>
> ## Weakness # 4: The choice of datasets in this study is limited to a single type.
> >
> > According to another recent work that uses realistic e-commerce datasets [10], the average user degree is usually less than 50, which aligns with the characteristics of the datasets we use in this work. Nevertheless, to verify that this phenomenon is also observable in dense datasets, we apply TAG-CF to Movielens-1M. It is a dense dataset where each user has an average number of 165 interactions, as opposed to ~50 interactions for other datasets that we utilize in our manuscript. The results are shown in the table below and we can notice that our observation regarding the performance improvement brought by message passing still holds.
>
> |  Method   |   MF  | +TAG-CF | TAG-CF's Improvement over MF | LightGCN | TAG-CF's Improvement over LightGCN |
> |:---------:|:-----:|:-------:|:----------------------------:|:--------:|:----------------------------------:|
> |           |       |         |     Low-degree Percentile    |          |                                    |
> |  NDCG@20  | 20.98 |  29.20  |             39.2%            |   25.95  |                12.5%               |
> | Recall@20 | 23.64 |  28.10  |             18.9%            |   25.80  |                8.9%                |
> |           |       |         |            Overall           |          |                                    |
> |  NDCG@20  | 22.51 |  29.65  |             31.7%            |   26.64  |                11.3%               |
> | Recall@20 | 25.79 |  28.40  |             10.1%            |   26.30  |                8.0%                |
>
> Table Re3: The performance comparison between LightGCN and TAG-CF on Movielens-1M.

---

> ### Author Response · Authors · 2023-11-18
> **Response to Reviewer uTpx [6/7]**
>
> ## Weakness #5: Despite the computational complexity, the improvements achieved with TAG-CF+ are not significantly greater than those of TAG-CF.
> >
> > The goal of TAG-CF+ is to maintain the performance of TAG-CF and meanwhile further avoid redundant computational over high-degree users. While not intended, TAG-CF+ still improves the recommendation performance of TAG-CF, because TAG-CF sometimes hurts the performance of high-degree users even with overall performance improvement.
>
> > **[The selection of low-degree nodes]**
> With respect to your concerns about the additional complex tasks entailed by TAG-CF+, we would like to highlight that the computational bottleneck of TAG-CF+ depends on the one-time message passing (which is very cheap already), instead of the selection of low-degree nodes. Specifically, TAG-CF+ only searches the selection of low-degree nodes over the validation set. We first conduct regular TAG-CF on the validation set and then select the degree threshold as a post-processing operation by re-using the result from TAG-CF on the validation set. This post-processing operation is as simple as a slicing operation, which finishes in milliseconds on commercial CPUs. In our appendix, we also reported the efficiency improvement of TAG-CF+ to TAG-CF. The overall running time improves by ~10%, including the selection of low-degree nodes.
>
> > **[Passing messages to nodes with lower degrees and searching for neighbors of low-degree nodes]**
> This portion of computational overhead is inevitable for every single CF method that utilizes knowledge from graphs. In order to conduct message passing of any sort, one is required to acquire node neighbors and pass messages between adjacent nodes. Message passing in existing graph-based CF methods repetitively conducts these operations for every node and every training iteration. TAG-CF significantly reduces the computational overheads and conducts these operations only once for all nodes. TAG-CF+ further improves and conducts these operations only once for low-degree nodes. Besides, TAG-CF and TAG-CF+ only require 1-hop neighbor information, unlike existing works that usually require neighbors from 2 to 3 neighbors. The price that TAG-CF and TAG-CF+ pay is literally the minimal price if one wants to incorporate message passing of any sort in a CF system.
>
> > To quantitatively support our claim, the table below shows the running time of each operation that you mention. We can see that the node selection only takes 1% of the additional running time brought by TAG-CF+ or 0.001% of the total running time, which is almost negligible.
>
> | Dataset/Operation | LightGCN | MF+TAG-CF | MF+TAG-CF+ (passing messages to nodes with lower degree searching for neighbors of low-degree nodes) | Node selection in TAG-CF+ (10 thresholds) | Node selection in TAG-CF+ (50 thresholds) |
> |:-----------------:|----------|:---------:|:----------------------------------------------------------------------------------------------------:|:-----------------------------------------:|:-----------------------------------------:|
> |       Anime       | 138.85   |   34.16   |                                                 34.14                                                |                1.70 x 1e-4                |                2.28 x 1e-4                |
> |     Yelp-2018     | 5.81     |    3.19   |                                                 3.18                                                 |                0.92 x 1e-4                |                1.31 x 1e-4                |
> |      Gowalla      | 13.27    |    7.76   |                                                 7.75                                                 |                1.15 x 1e-4                |                1.42 x 1e-4                |
> |    Amazon-Book    | 46.62    |   29.24   |                                                 29.23                                                |                1.38 x 1e-4                |                1.75 x 1e-4                |
>
> Table Re4: The running time(1 x 10^3 seconds) for LightGCN, TAG-CF, and TAG-CF+.

---

> ### Author Response · Authors · 2023-11-18
> **Response to Reviewer uTpx [7/7]**
>
> ## Questions # 1: Number of layers in section 3.2
>
> We conduct a hyper-parameter tuning over the selection of 2-3 layers and choose the setup with optimal performance. Specifically, on both Gowalla and Yelp-2018, we choose a two-layer LightGCN.
>
> ## Questions # 2: Experiments on datasets with even higher sparsity levels.
>
> We conduct experiments on our private internal dataset with extremely high sparsity (i.e., 99.99%). As for public benchmark datasets, I believe that Amazon-Book with a sparsity of 99.94% is sparse enough.
>
>
> **We hope we have satisfactorily answered your questions. If so, could you please consider increasing your rating? If you have remaining doubts or concerns, please let us know, and we will happily respond.**
>
> Reference: \
> [1] Wu, Jiancan, et al. "Self-supervised graph learning for recommendation." SIRIR 2021\
> [2] Yu, Junliang, et al. "Are graph augmentations necessary? simple graph contrastive learning for recommendation." SIGIR 2022\
> [3] Cai, Xuheng, et al. "LightGCL: Simple Yet Effective Graph Contrastive Learning for Recommendation." ICLR 2023\
> [4] He, Xiangnan, et al. "Lightgcn: Simplifying and powering graph convolution network for recommendation." SIGIR 2020\
> [5] Wu, Shiwen, et al. "Graph neural networks in recommender systems: a survey." CSUR 2022\
> [6] Kelong Mao, et al., “UltraGCN: Ultra Simplification of Graph Convolutional Networks for Recommendation” CIKM 2021\
> [7] Si, Si, et al. "Serving Graph Compression for Graph Neural Networks." ICLR 2023\
> [8] Zhang, Shichang, et al. "Graph-less neural networks: Teaching old mlps new tricks via distillation." ICLR 2022\
> [9] Guo, Zhichun, et al. "Linkless link prediction via relational distillation." ICML 2023\
> [10] Zheng, Wenqing, et al. "Cold brew: Distilling graph node representations with incomplete or missing neighborhoods." ICLR 2022

---

> ### Author Response · Authors · 2023-11-22
> **Reminder about Rebuttal.**
>
> Dear Reviewer uTpx,
>
> As today is the final day of our discussion, we anticipate the opportunity to engage with you. If you have any remaining questions or concerns, please don't hesitate to share them with us, and we will be happy to respond. Thank you.
>
> Best regards,\
> TAG-CF authors

---

### Official Review · Reviewer_bF6q · 2023-11-06

**Soundness:** 3 good
**Presentation:** 3 good
**Contribution:** 3 good
**Rating:** 8
**Confidence:** 4

**Summary:**

The paper discusses the (positive) role of message passing in graph collaborative filtering. The analysis is initially driven by the assumption that, even though message passing in graph collaborative filtering is applied exactly as it appears in other graph learning tasks, evidence (in terms of recommendation performance) demonstrates that message passing in graph collaborative filtering may be working in a different manner. On one side, through a simple reformulation of the message passing (in the case of LightGCN) the authors show that it inherently comes with the usual user-item similarity score (as in MF) plus additional inductive biases accounting for other more refined interactions between users and/or items. This suggests that the message passing could improve MF-like approaches in two ways, namely: 1) neighborhood aggregation and 2) the additional gradient updates. An empirical analysis demonstrates that it is the neighborhood aggregation to provide the highest contribution to the improved performance. On another side, the authors empirically and mathematically prove that, differently from what happens in graph learning, high degree nodes seem to benefit from message passing more than low degree ones. In the light of above, the authors propose TAG-CF, short for Test-time Aggregation for Collaborative Filtering, a simple but effective model agnostic solution which performs message passing on top of any MF-like recommender system only at inference time. Extensive experimental analyses confirm the efficacy of the proposed approach over several baselines and on popular recommendation datasets. The evaluation is complemented through ablation studies and a computational time assessment.

**Strengths:**

+ The paper proposes a pivotal question to assess the actual role of message passing in collaborative filtering.
+ The empirical and theoretical preliminary analyses are sound and help supporting the proposal of the TAG-CF solution.
+ The proposed approach is simple and effective from a theoretical and experimental point of view.
+ The experimental setting is extensive.
+ The code is released at review time.

**Weaknesses:**

- Some important related work and baselines may be missing. For instance, GFCF [1] is another work questioning the role of graph convolutional network in recommendation; UltraGCN [2] and SVD-GCN [3] discuss the role of additional neighborhood aggregation types (e.g., user-user and item-item) in collaborative filtering.
- Some clarification needs to be provided regarding the low degree aspect (i.e., section 3.2).

[1] Yifei Shen, Yongji Wu, Yao Zhang, Caihua Shan, Jun Zhang, Khaled B. Letaief, Dongsheng Li: How Powerful is Graph Convolution for Recommendation? CIKM 2021: 1619-1629

[2] Kelong Mao, Jieming Zhu, Xi Xiao, Biao Lu, Zhaowei Wang, Xiuqiang He: UltraGCN: Ultra Simplification of Graph Convolutional Networks for Recommendation. CIKM 2021: 1253-1262

[3] Shaowen Peng, Kazunari Sugiyama, Tsunenori Mine: SVD-GCN: A Simplified Graph Convolution Paradigm for Recommendation. CIKM 2022: 1625-1634

**After the rebuttal.** The rebuttal clarified all weaknesses.

**Questions:**

* Did the authors consider testing the proposed approach against UltraGCN? Indeed, UltraGCN is described as extremely simplified version of LightGCN also in terms of computational time. What is more, it almost removes the message passing from the training phase and proposes an approximation of infinite propagation layers through additional loss components.
* Reading the discussion about low and high degree nodes in section 3.2, it seems that the observed behaviour (i.e., good performance on high degree nodes) could be ascribed to the fact that, in general, all recommendation approaches built on the collaborative filtering paradigm tend to provide higher-quality recommendations for active users at the detriment of less active ones (i.e., warm/cold users respectively). Thus, maybe this trend is only linked to the specific characteristics of each recommendation dataset, and it is not unique for graph-based recommender systems. Could the authors elaborate on this aspect?

**After the rebuttal.** The rebuttal answered all questions.

---

> ### Author Response · Authors · 2023-11-18
> **Response to Reviewer bF6q [1/2]**
>
> Dear Reviewer bF6q:
>
> Thank you for your valuable feedback. We sincerely appreciate your acknowledgment of our paper’s theoretical soundness, practicality, and comprehensiveness for experiments. Our detailed response to your concerns is as follows:
>
> ## Question #1: Did the authors consider testing the proposed approach against UltraGCN?
> >
> > Thanks for pointing out these related works. We have modified our manuscript (i.e., in the experiment and related work sections with edits in blue) and accordingly added discussions over these methods (GFCF [1], UltraGCN [2], and SVD-GCN [3]). Following your suggestions, we listed UltraGCN as one of our baselines. The performance and efficiency comparison between TAG-CF and UltraGCN is shown in the table below.
>
> > From Table Re1, we observe that UltraGCN outperforms MF-BPR and MF-BPR+TAG-CF. However, compared with MF and TAG-CF from DirectAU, the performance of UltraGCN is not as competitive. Besides, we also apply TAG-CF to user/item representations trained by UltraGCN and we notice that TAG-CF can further enhance its promising performance (i.e., an average 6.1% improvement on NDCG@20 on these four datasets and 13.4% on Recall@20). While UltraGCN implicitly approximates the utilities of message passing through a series of additional regularization terms, these results indicate that TAG-CF can still improve performance by explicitly aggregating neighbor information.
>
> > Moreover, we compare the total running time of TAG-CF and UltraGCN. We notice that UltraGCN runs faster than LightGCN.  However, compared with vanilla MF, UltraGCN introduces a lot of computational overhead by repetitively calculating additional loss terms guided by the graph structure. On the other hand, TAG-CF consistently brings negligible overheads (i.e., less than 1% of the total running time).
>
> |    Method   |   MF-BPR   | MF-BPR + TAG-CF |    MF-DirectAU    | MF-DirectAU + TAG-CF |  UltraGCN  | UltraGCN + TAG-CF |
> |:-----------:|:----------:|:---------------:|:-----------------:|:--------------------:|:----------:|:-----------------:|
> |             |            |                 |  Overall NDCG@20  |                      |            |                   |
> | Amazon-Book |  4.15±0.13 |    4.32±0.13    |     8.01±0.03     |       8.13±0.03      |  5.77±0.25 |     6.11±0.27     |
> |    Anime    | 29.51±0.21 |    30.23±0.26   |     24.01±0.06    |      27.25±0.03      | 30.30±0.11 |     30.89±0.11    |
> |   Gowalla   |  7.51±0.12 |    7.99±0.14    |     9.77±0.08     |       9.88±0.04      |  8.53±0.14 |     9.02±0.15     |
> |  Yelp-2018  |  3.96±0.14 |    4.26±0.17    |     6.25±0.06     |       6.36±0.03      |  5.01±0.11 |     5.53±0.11     |
> |             |            |                 | Overall Recall@20 |                      |            |                   |
> | Amazon-Book |  7.35±0.22 |    7.64±0.20    |     12.67±0.06    |      12.97±0.06      |  8.01±0.25 |     8.53±0.25     |
> |    Anime    | 34.84±0.30 |    35.23±0.34   |     29.15±0.09    |      31.95±0.05      | 35.87±0.39 |     37.01±0.39    |
> |   Gowalla   | 14.47±0.23 |    14.92±0.25   |     18.30±0.17    |      18.53±0.11      | 15.93±0.21 |     16.36±0.22    |
> |  Yelp-2018  |  7.27±0.27 |    7.62±0.22    |     10.81±0.10    |      11.21±0.09      |  8.41±0.19 |     9.89±0.20     |
>
> Table Re1: The performance (NDCG@20 and Recall@20) of UltraGCN and TAG-CF.
>
> |   Method    | LightGCN |   MF  | +TAG-CF | UltraGCN | +TAG-CF |
> |:-----------:|:--------:|:-----:|:-------:|:--------:|:-------:|
> |    Anime    |  138.85  | 34.12 |  +0.04  |   93.31  |  +0.04  |
> |  Yelp-2018  |   5.81   |  3.17 |  +0.02  |   5.02   |  +0.02  |
> |   Gowalla   |   13.27  |  7.74 |  +0.02  |   12.55  |  +0.02  |
> | Amazon-Book |   11.54  | 29.21 |  +0.03  |   39.26  |  +0.03  |
>
> Table Re2: The running time comparison (1 x 10^3 seconds) between TAG-CF and UltraGCN.

---

> ### Author Response · Authors · 2023-11-18
> **Response to Reviewer bF6q [2/2]**
>
> ## Questions 2:  Reading the discussion about low and high degree nodes in section 3.2.
> > **[Phenomenon w.r.t. the user degree]** Thanks for your insightful comments. We agree with you on the idea that CF methods (graph-based or non-graph-based) give better recommendation results for active (i.e., high-degree) users than less active (i.e., low-degree) ones. We believe that this claim aligns with the argument we present in our manuscript. In our manuscript, we connect CF objective functions (i.e., BPR and DirectAU) to message passing and show that these CF objectives inadvertently conduct message passing during the back-propagation.
>
> > Since this inadvertent message passing happens during the back-propagation, its performance is positively correlated to the amount of training signals a user/item can get. In the case of CF, the amount of training signals for a user is directly proportional to the node degree of this user. High-degree active users naturally benefit more from the inadvertent message passing from objective functions like BPR and DirectAU, because they acquire more training signals from the objective function. Hence, when explicit message passing is applied to CF methods, the performance gain for high-degree users is less significant than that for low-degree users. Because the contribution of the message passing over high-degree nodes has been mostly fulfilled by the inadvertent message passing during the training.
>
> > We would like to emphasize that our study does not only show that graph-based and non-graph-based CF methods deliver better performance for active high-degree nodes; this observation is already well-known in the community, as you have indicated. However, the most significant observation that we would like to highlight is that the explicit incorporation of message passing (e.g., LightGCN vs. MF, or TAG-CF vs. MF) helps low-degree users more, compared with high-degree users. We believe that this novel observation is unique to graph-based CF methods and valuable to the recommender system community.
>
> > **[Dataset-specific]**
> We believe this is not a dataset-specific phenomenon; instead, it is a phenomenon commonly seen in the recommender systems community. Almost every dataset in the recommender system community has a heavy-tailed distribution w.r.t. the user degree, where both active high-degree users and less active low-degree users co-exist. Hence, even in dense datasets where the average interaction per user is higher, high-degree users still receive substantially more training signals than low-degree users. So the observations and claims we made in this work are broadly applicable to message passing in CF across a broad range of datasets.
>
> > According to another recent work that uses realistic e-commerce datasets [4], the average user degree is usually less than 50, which aligns with the characteristics of the datasets we use in this work. Nevertheless, to verify that this phenomenon is also observable in dense datasets, we apply TAG-CF to Movielens-1M. It is a dense dataset where each user has an average number of 165 interactions, as opposed to ~50 interactions for other datasets that we utilize in our manuscript. The results are shown in the table below and we can notice that our observation regarding the performance improvement brought by message passing still holds.
>
> |  Method   |   MF  | +TAG-CF | TAG-CF's Improvement over MF | LightGCN | TAG-CF's Improvement over LightGCN |
> |:---------:|:-----:|:-------:|:----------------------------:|:--------:|:----------------------------------:|
> |           |       |         |     Low-degree Percentile    |          |                                    |
> |  NDCG@20  | 20.98 |  29.20  |             39.2%            |   25.95  |                12.5%               |
> | Recall@20 | 23.64 |  28.10  |             18.9%            |   25.80  |                8.9%                |
> |           |       |         |            Overall           |          |                                    |
> |  NDCG@20  | 22.51 |  29.65  |             31.7%            |   26.64  |                11.3%               |
> | Recall@20 | 25.79 |  28.40  |             10.1%            |   26.30  |                8.0%                |
>
> Table Re3: The performance comparison between LightGCN and TAG-CF on Movielens-1M.
>
> **In light of our answers to your concerns, we hope you consider raising your score. If you have any more concerns, please do not hesitate to ask and we'll be happy to respond.**
>
> Reference:\
> [1] Yifei Shen, et al., “How Powerful is Graph Convolution for Recommendation?” CIKM 2021 \
> [2] Kelong Mao, et al., “UltraGCN: Ultra Simplification of Graph Convolutional Networks for Recommendation” CIKM 2021\
> [3] Shaowen Peng, et al., “SVD-GCN: A Simplified Graph Convolution Paradigm for Recommendation”. CIKM 2022\
> [4] Zheng, Wenqing, et al. "Cold brew: Distilling graph node representations with incomplete or missing neighborhoods." ICLR 22

---

> > ### Comment · Reviewer_bF6q · 2023-11-20
> >
> > Dear Authors,
> >
> > thank you for your careful rebuttal. I'll answer to both points you outlined.
> >
> > **Question 1.** Thank you for providing additional experiments and results with UltraGCN. They further confirm your analysis and the proposal of the TAG-CF approach.
> >
> > **Question 2.**  Your response is extensive and helped me solving most of the previous doubts regarding the high- and low-degree nodes.
> >
> > Your rebuttal did answer to all my concerns, and convinced me even more about the initial rating I gave to your work.

---

> > > ### Author Response · Authors · 2023-11-20
> > > **Thanks for your reply.**
> > >
> > > Dear Reviewer bF6q:
> > >
> > > We greatly appreciate your valuable insights and constructive feedback on our paper. It's truly encouraging to see that our response effectively addresses the concerns you raised. Thank you for recognizing the efforts we've put into our work.
> > >
> > > Best regards,\
> > > TAG-CF authors

---

### Author Response · Authors · 2023-11-18
**General Response to All Reviewers**

We thank the reviewers for their feedback and constructive suggestions. We are pleased that reviewers unanimously acknowledged the impact of our research, said “proposes a pivotal question” (bF6q), “promising avenue of investigation” (uTpx), “demonstrate the usability of TAG-CF” (gGxd), and “study is reasonable and novel” (6tQK).

At the same time, multiple reviewers expressed concerns about our conclusions regarding. the effectiveness of message passing to low-degree and high-degree users. Summarizing suggestions from all reviewers, from the perspective of the number of supervision signals, we conduct additional experiments on upsampling the low-degree users. Though upsampling low-degree users hurts the overall performance, we notice that the performance improvement brought by TAG-CF for low-degree users decreases, as the upsampling rate increases. We can conclude that the more supervision signals a user receives (no matter for a low-degree or high-degree user), the less performance improvement message passing can bring. These experiments quantitatively show why the performance improvement of high-degree users could be limited more than low-degree users. Because high-degree users naturally receive more training signals during the training whereas low-degree users receive fewer training signals.

Besides, while performance improvement to state-of-the-art models is critical for driving progress in the field, efficient and effective simplifications to SoTA are also essential for advancing our collective understanding of the existing research gaps.

An important contribution of our work is to analyze how the most popular message passing scheme (i.e., linear neighbor aggregation) in CF improves the recommendation performance. By conducting ablation studies over the message passing scheme, we empirically demonstrated that the message passing in CF does not have to be applied during the model training (i.e., Section 3.1). Furthermore, by connecting CF objective functions (i.e., BPR and DirectAU) to message passing, we showed that the performance improvement on high-degree users could be limited more than low-degree users (i.e., Section 3.2). Hence message passing on high-degree nodes could be dropped to reduce the overall computational overhead. Leveraging these novel takeaways, we propose TAG-CF and TAG-CF+ that effectively enhance MF methods by conducting message passing only once at test time. They are extremely flexible, simple to implement and enjoy the performance benefits of the graph-based CF method while paying the lowest overall scalability.

While end-to-end trained graph-based CF methods consistently deliver promising performance, they are rarely used in real-world production scenarios, due to training complexities and prohibitively expensive overheads. TAG-CF offers an extremely simple yet effective and practical approach for industry settings to benefit from the power of message passing while paying minimal cost. We believe that the theoretical and empirical findings we provide in this paper will be valuable to researchers from both academia and industry.

We also improved our manuscript by leveraging valuable comments from the reviewers. The modifications (i.e., edits in blue) we made are listed as follows:

* With respect to the comments and suggestions regarding. low-degree and high-degree users’ performance, we added additional clarifications in Section 3.2 to elaborate on why the connection between objective functions and message passing leads to the discrepancy between the performance improvement on low- and high-degree users.

* With respect to the comments on experiments over additional baselines and datasets, we added an extra section in the appendix (i.e., Appendix H on page 19) to fully demonstrate the effectiveness and legitimacy of our findings and proposal.

* With respect to the comments on related works, we added extra experiments and discussions over graph-based CF methods that also aim at accelerating and simplifying the message passing in CF.

**We modified the manuscript (i.e., edits in blue) to clarify your concerns, and we hope we have satisfactorily answered your questions. If so, could you please consider increasing your rating? If you have remaining doubts or concerns, please let us know, and we will happily respond. Thank you!**

Best regards,\
TAG-CF authors

---

> ### Comment · Reviewer_bF6q · 2023-11-20
>
> Dear Authors,
>
> besides the rebuttal to my review, I also read the other reviews-rebuttals. The answers you gave in the rebuttals were, overall, very careful and well-structured.
>
> As already stated in my rebuttal, this convinces me even more about the initial acceptance rating I gave to your work.
>
> Good luck with the other rebuttals!

---

> ### Author Response · Authors · 2023-11-20
> **Thanks for your reply.**
>
> Dear Reviewer bF6q:
>
> Your valuable insights and constructive feedback on our paper are deeply appreciated. It is motivating to see that our responses have successfully resolved your concerns. We are grateful for being acknowledged at the dedication we've invested in our research. We will actively engage in the discussion with other reviewers and refine our submission accordingly as much as possible.
>
> Best regards,\
> TAG-CF authors

---

### Author Response · Authors · 2023-11-22
**Reminder about Rebuttal**

Dear Reviewers and Chairs,

We sincerely appreciate your constructive feedbacks and efforts in helping us to strengthen the paper. By leveraging your valuable comments, we accordingly modified our manuscript (i.e., edits in blue). While the discussion phase is approaching to an end, if there is any final-round question or suggestion with respect to our work, we are always open to discussion.

We hope we have satisfactorily answered your questions. If so, could you please consider increasing your rating? If you have remaining doubts and concerns, please let us know, and we will happily respond. We would like to thank you again for your great efforts in reviewing our paper.

Best regards,\
TAG-CF authors

---

### Meta-Review · Area_Chair_pWzV · 2023-12-11

**Metareview:**

This paper studies the effectiveness of message passing in CF and challenges previous assumptions. Through rigorous ablation studies, it is discovered that message passing primarily improves CF performance through information passed from neighbors rather than their gradients, and it tends to benefit low-degree nodes more than high-degree nodes. Based on these findings, a test-time augmentation framework called TAG-CF is proposed, which conducts message passing once at inference time and effectively enhances representations trained by different CF supervision signals. There are some weaknesses of the paper raised from the review comments and discussions, including the inadequate analysis of the experiments and theory, the significance of the results, and the presentation. Although the authors provided detailed feedbacks, some of the concerns raised are still unsolved.

**Justification For Why Not Higher Score:**

There are some weaknesses of the paper raised from the review comments and discussions, including the inadequate analysis of the experiments and theory, the significance of the results, and the presentation. Although the authors provided detailed feedbacks, some of the concerns raised are still unsolved.

**Justification For Why Not Lower Score:**

N/A

---

### Decision · Program_Chairs · 2024-01-16

Reject